# The alternative regenerative strategy of bearded dragon unveils the key processes underlying vertebrate tooth renewal

**Lotta Salomies, Julia Eymann, Imran Khan, Nicolas Di-Poï\***

Program in Developmental Biology, Institute of Biotechnology, University of Helsinki, Helsinki, Finland

**Abstract** Deep understanding of tooth regeneration is hampered by the lack of lifelong replacing oral dentition in most conventional models. Here, we show that the bearded dragon, one of the rare vertebrate species with both polyphyodont and monophyodont teeth, constitutes a key model for filling this gap, allowing direct comparison of extreme dentition types. Our developmental and high-throughput transcriptomic data of microdissected dental cells unveils the critical importance of successional dental lamina patterning, in addition to maintenance, for vertebrate tooth renewal. This patterning process happens at various levels, including directional growth but also gene expression levels, dynamics, and regionalization, and involves a large number of yet uncharacterized dental genes. Furthermore, the alternative renewal mechanism of bearded dragon dentition, with dual location of slow-cycling cells, demonstrates the importance of cell migration and functional specialization of putative epithelial stem/progenitor niches in tissue regeneration, while expanding the diversity of dental replacement strategies in vertebrates.
DOI: https://doi.org/10.7554/eLife.47702.001

## Introduction

Exposed to the oral cavity of most jawed vertebrates, teeth are continually worn down through food acquisition, a process that must be compensated to maintain the oral apparatus functional (*Berkovitz and Shellis, 2016*). The most common, and likely ancestral, solution to this problem is polyphyodonty, or continuous replacement of the dentition, through shedding of functional teeth (*Jernvall and Thesleff, 2012*; *Tucker and Fraser, 2014*). Whereas polyphyodonty prevails in the majority of vertebrate taxa, mammals show a substantial reduction in the number of tooth generations, a phenomenon often associated with increased dental complexity in this group (*Kielan-Jaworowska et al., 2004*). Particularly, the vast majority of mammals display a diphyodont dentition, producing only two sets of teeth—deciduous or 'milk' teeth and permanent replacement teeth. Rodents such as *Mus musculus*, one of the most commonly used vertebrate animal models, are even more extreme and develop only one generation of teeth, a condition known as monophyodonty (*Jernvall and Thesleff, 2012*). As a result, although mouse developmental studies of the past decades have revealed many of the signaling pathways controlling primary tooth development, relatively little is known about the molecular mechanisms underlying lifelong tooth regeneration in vertebrates.

Recent studies in polyphyodont and diphyodont species from all major vertebrate groups have shown that, with the exception of some teleost fish such as salmonids (*Fraser et al., 2006*; *Jernvall and Thesleff, 2012*; *Vandenplas et al., 2016*), tooth replacement is generally dependent on a specialized epithelial structure, the dental lamina (DL). This structure first emerges as a sheet of epithelial cells that invaginates from the oral epithelium (OE), and plays a crucial role in the developmental events contributing to primary tooth formation. The DL is maintained throughout the lifetime

**\*For correspondence:**
nicolas.di-poi@helsinki.fi

**Competing interests:** The authors declare that no competing interests exist.

**eLife digest** All multicellular organisms, from lizards to humans, must be able to repair and regrow damaged tissue. This includes not only healing after an injury, but also replacing parts of the body that suffer wear and tear. For example, many animals shed and replace worn out teeth throughout their life, but the number of times this occurs varies greatly between species.

Much of the understanding about how teeth grow and develop has come from researching mice. However, mice only develop one set of teeth, making them a poor 'model' for studying how species such as fish and reptiles can re-grow and replace their teeth. Recent studies of these species has shown that regenerating teeth relies on a specialised structure known as the dental lamina. In mice, the dental lamina forms but then quickly disappears, preventing new sets of teeth from developing. In most animals that regrow their teeth, however, the dental lamina keeps growing beyond the most recently produced tooth to create an area where its replacement will emerge. Now, Salomies et al. have identified other strategies involved in tooth replacement from studying the bearded dragon lizard, a rare example of an animal that continuously regenerates some, but not all, of its teeth.

Analysing the cells in different parts of the re-growing teeth from bearded dragon lizards revealed new features of the dental lamina. Specifically, Salomies et al. found that a previously uncharacterized set of genes within the dental lamina could determine whether or not a tooth will be replaced. Further experiments using microscope imaging revealed that bearded dragon lizards use two distinct groups of stem cells – specialised cells that have the potential to develop into various cell types in the body – to re-grow their teeth. These experiments demonstrate how the bearded dragon lizard uses a previously unknown mechanism to regenerate its teeth, combining elements used by gecko lizards and sharks.

These findings are an important step towards understanding the different strategies animals can use to maintain and regenerate their teeth. The knowledge gained could one day help design better therapies for patients suffering from inherited dental disorders or tooth loss.
DOI: https://doi.org/10.7554/eLife.47702.002

of polyphyodont species, and continues to grow beyond the previously generated tooth to create a free margin, the successional dental lamina (SDL), which constitutes the initiation site of replacement teeth. In monophyodont or diphyodont species such as mouse (*Dosedělová et al., 2015*), minipig (*Buchtová et al., 2012*), ferret (*Jussila et al., 2014*), and chameleon (*Buchtová et al., 2013*), both the DL and associated SDL degenerate relatively rapidly following formation of either one or two sets of teeth, thus highlighting the importance of maintaining these epithelial structures for tooth replacement. Coherent with this, the DL is considered to be the main source of putative odontogenic stem cells required for long-term tooth renewal (*Huysseune and Thesleff, 2004*; *Smith et al., 2009*; *Jernvall and Thesleff, 2012*; *Juuri et al., 2013*; *Thiery et al., 2017*), and label-retaining cells (LRCs) have been initially mapped to this region in the polyphyodont leopard gecko (*Handrigan et al., 2010*). More recently, putative stem/progenitor populations have been identified in the distal DL bulge of the American alligator (*Wu et al., 2013*), in the SDL and OE of sharks (*Martin et al., 2016*), as well as in the superficial DL of pufferfish (*Thiery et al., 2017*), thus indicating variations in dental regenerative strategies among polyphyodont species. Molecularly, compelling evidence from a number of reptilian species, teleost fish, shark, and ferret has revealed a conserved set of transcription factors and signaling molecules exhibiting similar expression patterns in the dental apparatus. In particular, the canonical Wnt pathway has emerged as a strong candidate signaling pathway associated with tooth replacement, because of its local activation (as revealed by the expression of Wnt/β-catenin target genes) in most studied polyphyodont and diphyodont species (*Handrigan and Richman, 2010a*; *Fraser et al., 2013*; *Gaete and Tucker, 2013*; *Wu et al., 2013*; *Jussila et al., 2014*; *Rasch et al., 2016*), but also because of its capacity to produce a large number of supernumerary teeth when activated in snakes and transgenic mice (*Järvinen et al., 2006*; *Wang et al., 2009*; *Gaete and Tucker, 2013*). Furthermore, stabilizing Wnt/β-catenin signaling was recently shown to induce formation of replacement tooth germs in the normally degenerating mouse SDL, thus providing further support for the important role of this pathway in controlling tooth induction and regeneration (*Popa et al., 2019*). However, only low-scale, single- or cross-

species candidate gene approaches based on mouse expression patterns have been used so far to characterize polyphyodont dentitions, and the complete molecular signature of vertebrate tooth regeneration is still unknown.

In this study, we examined one agamid lizard model, the bearded dragon (*Pogona vitticeps*), which offers an excellent model system to study the events that define tooth development and regeneration because of its uncommon dentition with distinct modes of implantation and regenerative capacities along the same jaw—acrodont, monophyodont teeth posteriorly and pleurodont, polyphyodont teeth anteriorly (*Cooper et al., 1970*; *Di-Poï and Milinkovitch, 2016*). This lizard is one of the rare extant vertebrate species with such marked heterodonty, which allows direct comparisons of extreme types of vertebrate dentition in a single model. Unlike the dentition of other polyphyodont reptiles investigated so far, our data further indicate that bearded dragons possess a relatively slow replacement process (based on a 'one-for-one' strategy) as well as a permanent, proliferating SDL that only initiates replacement at anterior polyphyodont locations, making this model extremely attractive for elucidating the key importance of DL and SDL epithelial structures in vertebrate tooth initiation and regeneration. Indeed, we show here that the early degradation of the SDL is not the only mechanism restricting vertebrate tooth replacement (*Buchtová et al., 2012*; *Dosedělová et al., 2015*). Instead, our developmental data and large-scale transcriptomic analysis of microdissected dental tissues unveil the critical importance of SDL patterning at various levels, including directional growth but also gene expression levels, dynamics, and regionalization, for tooth regeneration. Particularly, we identify a large number of yet uncharacterized developmental genes associated with dental patterning in regenerating and non-regenerating teeth, so our lizard model is also uniquely suited for revealing new signaling pathways critical for tooth replacement in vertebrates. Additionally, our bearded dragon findings reveal a novel mechanism of continuous tooth replacement in vertebrates, which shares with the leopard gecko a source of putative odontogenic progenitor/stem cells in the DL but closely parallels the shark by the presence of a separate second source of cells originating from the OE. This novel dental replacement strategy definitely demonstrates that the bearded dragon provides a rare opportunity to expand our understanding of tooth developmental dynamics and diversity, while providing a unique model system to study the function and specialization of epithelial structures and dental progenitor/stem cell niches present in distinct groups of vertebrates. Altogether, and given the well-established history in captivity and key advantages of the bearded dragon lizard for research (*Ollonen et al., 2018*) this emerging model organism offers a powerful system to elucidate the developmental and genetic basis of both evolutionary novelty and tooth regeneration, thus allowing to answer both developmental and evolutionary questions critical for the design of more appropriate therapies targeted at congenital dental disorders and tooth loss in humans.

## Results

### Dental morphology and odontogenesis in bearded dragon

Agamid lizards possess an atypical heterodont dentition with both acrodont, monophyodont and pleurodont, polyphyodont tooth types, which vary widely in terms of number and shape among species and over ontogeny (*Cooper et al., 1970*) and personal observations). We first examined the gross anatomy of the dentition and associated DL in the bearded dragon, using micro-computed tomography (CT) imaging. Hatchling animals possess seven posterior triangular acrodont teeth attached to the jaw bones on each jaw quadrant, whereas the most anterior tooth positions on the premaxilla (three teeth, upper jaw) and dentary (two teeth, lower jaw) are occupied by pleurodont teeth (*Figure 1A,B*). At this early stage, the central-most tooth projecting forwards in the midline of the premaxilla consists of the large egg-tooth, which is later replaced by a regular pleurodont successor (*Figure 1A*). To visualize oral soft tissue, including the DL structure, we stained the jaws of juvenile bearded dragons with the contrast agent phosphotungstic acid (PTA) before CT-scanning. As expected, the DL forms a continuous sheet of epithelium spanning the whole jaws, without clear separation between the pleurodont and acrodont dentitions (*Figure 1C,D*). However, when compared to regions occupied by pleurodont teeth, the DL is shorter both at the level of acrodont teeth and in interdental regions, a morphology that is in line with the absence of tooth replacement at these particular positions (*Figure 1D*).

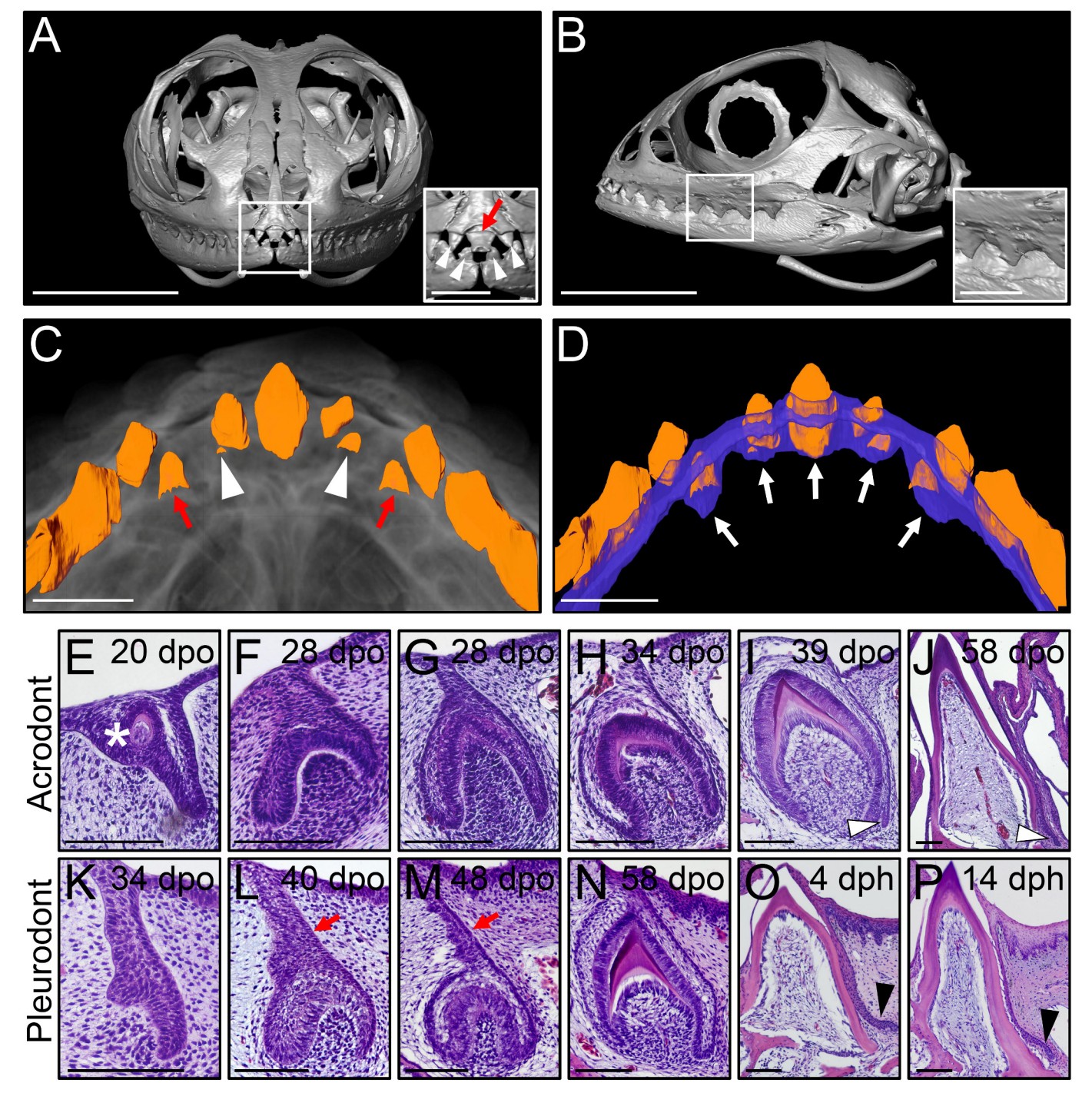

**Figure 1.** Different types of bearded dragon teeth develop through morphologically equivalent stages. (A, B) Micro-CT scan images of bearded dragon skull at hatchling stage in frontal (A) and lateral (B) views. Insets show high magnifications of anterior pleurodont (A) and posterior acrodont (B) teeth. Hatchling bearded dragons possess four pleurodont teeth (one in each jaw quadrant; white arrowheads) and one central egg tooth (red arrow). Scale bars: 1 cm (main images); 2.5 mm (insets). (C, D) 3D volume rendering as well as tooth (C, D; orange) and dental lamina (DL) (D); blue) segmentation of PTA-stained juvenile bearded dragon upper jaw in ventral views, showing the morphogenesis (C), red arrows) or replacement (C), white arrowheads) of anterior pleurodont teeth. Only the segmented teeth and DL are shown in panel (D) to highlight the expansion of the DL along the oral-aboral axis at pleurodont tooth positions (white arrows). Scale bars: 1 mm. (E–P) Hematoxylin and eosin (H and E)-stained coronal (E–J) or sagittal (K–P) sections of developing acrodont (E–J) and pleurodont (K–P) teeth at various developmental stages indicated as embryonic days post-oviposition (dpo) or days post-hatching (dph). Vestigial teeth develop directly from the oral epithelium (OE) prior emergence of functional dentition (E;

*Figure 1 continued on next page*

*Figure 1 continued*

asterisk). Pleurodont teeth emerge later in development than acrodont teeth, and the DL connecting the pleurodont tooth germs to the OE surface is extended (**L, M**; red arrows). The acrodont successional dental lamina (SDL) starts to develop at late mineralization stage and persists until hatching (**I, J**; white arrowheads), while the pleurodont SDL is only visible around hatching time (**O, P**; black arrowhead). Scale bars: 100 µm.

DOI: https://doi.org/10.7554/eLife.47702.003

We next assessed ontogenetic changes in bearded dragon tooth development, by comparing the post-ovipositional embryonic morphogenesis of the two dentition types with histological stainings (*Figure 1E–N*). As already shown in a previous study focusing on primary tooth morphogenesis (*Handrigan and Richman, 2010b*), odontogenesis at acrodont positions starts around 15 days post-oviposition (dpo) with the emergence of non-functional vestigial teeth developing directly from the OE (*Figure 1E*). These superficial teeth abort morphogenesis prematurely following early mineralization, and functional acrodont teeth subsequently develop through classical stages of odontogenesis from a primary DL invaginating from the OE (*Figure 1E–J*), as already described in a number of reptilian species (*Handrigan and Richman, 2010a*; *Buchtová et al., 2013*). Around the time of tooth eruption, a small SDL emerges on the lingual side of acrodont teeth, but in contrast to previous reptile observations (*Richman and Handrigan, 2011*; *Buchtová et al., 2013*), our data indicate that the SDL is not rudimentary but rather continues growing in size until hatching (*Figure 1I–J*) and even afterwards at postnatal stages (see *Figure 2B,C*). Importantly, whereas the developmental stages are morphologically equivalent for the whole dentition, our direct comparisons of acrodont and pleurodont teeth clearly indicate a developmental delay in the initiation of pleurodont teeth, which only occurs around 40 dpo. Furthermore, despite the overall morphological similarities between the two types of teeth, pleurodont teeth show an elongated primary DL and thus develop deeper inside the mesenchyme than acrodont teeth (*Figure 1L*). Finally, a different directional growth of the SDL is also evident, with pleurodont SDL growing lingually to accommodate the replacement tooth into the jaw, and acrodont SDL projecting downwards towards the jaw bone (*Figure 1J* and *Figure 2*).

## Gene expression patterns in embryonic and postnatal teeth

To investigate odontogenesis at the molecular level, we first inspected the expression pattern of conserved dental genes known to be associated with DL and SDL structures (*SHH*, *PITX2*, *NOTCH1*, and *LEF1*) by in situ hybridization (ISH), using various embryonic developmental stages in acrodont teeth: 'vestigial tooth/DL' stage (24 dpo), 'bell' stage (28 dpo), and 'mineralization/SDL development' stage (48 dpo; *Figure 2A*). The bearded dragon expression pattern of *SHH* and *PITX2* is comparable to previous reports in polyphyodont species (*Fraser et al., 2008*; *Handrigan and Richman, 2010b*; *Jussila and Thesleff, 2012*; *Wu et al., 2013*; *Rasch et al., 2016*). *PITX2* marks the primary DL at early stages, then becomes strongly expressed throughout the enamel organ and DL in both vestigial and bell stage teeth, and finally remains in the cervical loops, odontoblasts, DL, and SDL at mineralization stage. In contrast, *SHH* localizes specifically to the inner enamel epithelium of vestigial teeth and functional teeth, moving downwards towards the cervical loops at later stages, but no expression is detected in the SDL (*Figure 2A*). Notch signaling regulates stem/progenitor activity in several tissues and is particularly required for both tooth morphogenesis and maintenance of murine incisor stem cells (*Mitsiadis et al., 2005*; *Felszeghy et al., 2010*). Similarly to the mouse dentition, *NOTCH1* is predominantly expressed in the epithelial compartment of functional teeth, including in the DL tip at 24 dpo and OE at both 24 and 28 dpo, but its expression is also noticeable in the condensed mesenchyme of vestigial teeth at early developmental stages. Finally, our analysis of the Wnt readout gene *LEF1* indicates a condensed expression in the primary DL and at the tip of the SDL, a pattern previously observed in other squamate species (*Richman and Handrigan, 2011*; *Gaete and Tucker, 2013*) but contrasting with previous reports in bearded dragon acrodont SDL (*Richman and Handrigan, 2011*). Indeed, consistent with the equivalent morphology and persistence of SDL in both acrodont and pleurodont dentitions (*Figure 1J,N*), all tested genes were expressed in a similar pattern in both types of embryonic teeth (*Figure 2* and data not shown). Even further, our investigation of postnatal animals indicates the persistence of a highly proliferative SDL in both acrodont and pleurodont teeth at juvenile and adult stages (*Figure 2B,C*), as assessed by detection of proliferating nuclear cell antigen (PCNA) proliferation markers and BrdU incorporation. Additionally, in contrast

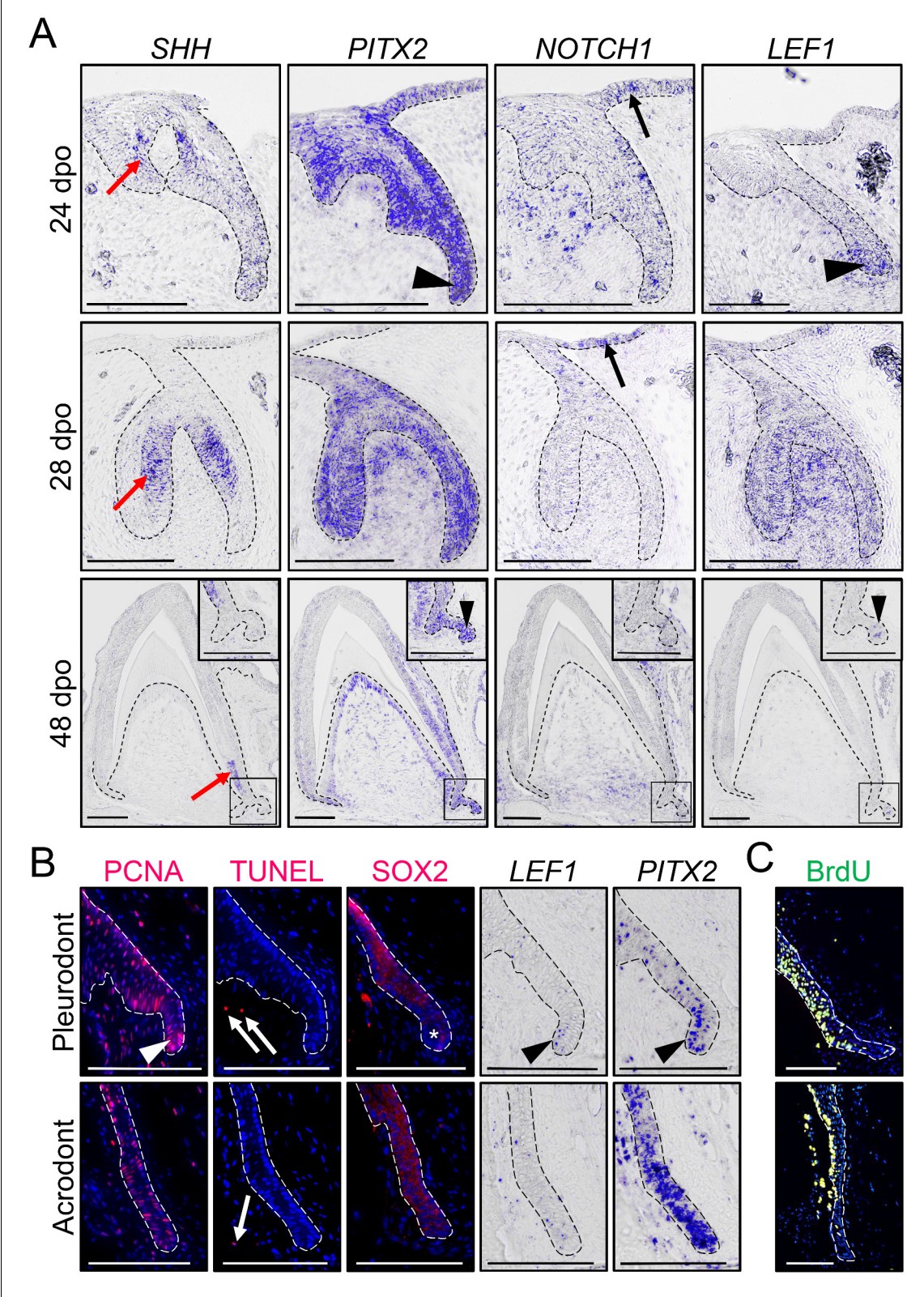

**Figure 2.** Expression of conserved dental genes is maintained from embryonic development to replacement. (**A**) In situ hybridization (ISH) showing the expression of *SHH*, *PITX2*, *NOTCH1*, and *LEF1* at various indicated developmental stages (24, 28, and 48 dpo) in developing mandibular acrodont teeth. The positive signal is false-colored to enhance visibility. Insets at 48 dpo show high magnifications of the SDL region indicated by black outlines. Black arrowheads indicate the expression of *PITX2* and *LEF1* in both the primary DL (24 dpo) and SDL (48 dpo). Black arrows indicate the expression of

*Figure 2 continued on next page*

Figure 2 continued

NOTCH1 in the OE, and red arrows show the expression of SHH in the inner enamel epithelium of both vestigial and functional teeth. (B) PCNA immunohistochemistry (IHC) and TUNEL apoptotic assay (left panels; red staining), SOX2 IHC (middle panel; red staining), or PITX2 and LEF1 ISH (right panels) in parallel sections of pleurodont (top panels) and acrodont (bottom) SDL in juvenile bearded dragon (<1 year old). Arrowheads and asterisks indicate focal or absent expression in the SDL tip of pleurodont teeth, respectively. Arrows show positive apoptotic cells in mesenchymal tissues. (C) BrdU IHC (green) in pleurodont (top panel) and acrodont (bottom) SDL in adult bearded dragon (>2 years old) following a 7 day BrdU pulse. The epithelium-mesenchyme junction is indicated by black or white dashed lines in all panels (A–C), and cell nuclei are counterstained with DAPI (blue staining) in IHC and TUNEL experiments (B, C). Scale bars: 100 μm.

DOI: https://doi.org/10.7554/eLife.47702.004

to juvenile chameleon lizards (*Buchtová et al., 2013*), no signs of SDL deterioration or increased apoptosis were observed in the acrodont SDL (*Figure 2B*). However, whereas PCNA, *LEF1*, and *PITX2* remain expressed in both acrodont and pleurodont teeth, careful examination of these markers indicates both oral-aboral and labial-lingual asymmetry in the pleurodont SDL, in contrast to the scattered pattern in acrodont SDL (*Figure 2B*). A complementary pattern was observed by immunohistochemistry (IHC) for SOX2, which is excluded from the very tip of pleurodont DL (*Figure 2B*), as already observed in some other regenerating species (*Juuri et al., 2013*; *Popa et al., 2019*). Altogether, our data indicate that despite the lack of initiation of replacement teeth at acrodont positions in bearded dragon, the SDL does not degenerate and continues growing in size into adulthood. To our knowledge, this is the first example of long-term SDL maintenance in non-polyphyodont dentition, suggesting that this process is independent of and not causally related to tooth replacement. Furthermore, the observed differences in spatial organization of cell proliferation and developmental gene expression between postnatal acrodont and pleurodont dentitions likely reflect the need of directional growth and focal gene expression foreshadowing initiation of pleurodont tooth replacement.

## Putative stem/progenitor cells in bearded dragon dentition

To assess the existence and location of putative slow-cycling stem/progenitor cells in bearded dragon teeth, we performed BrdU pulse-chase assays in juvenile animals. BrdU was administered for seven days (pulse), and samples were obtained after 0, 28, and 58 days of chase time. To facilitate identification of putative stem/progenitor cells in both acrodont and pleurodont teeth, we distinguished cells that have stopped proliferating (post-mitotic or terminally differentiated cells) from cells that are still cycling by the use of BrdU immunostaining combined with PCNA detection at each time point (*Figure 3A*). In addition, the relatively long half-life of PCNA increases the likelihood of correctly identifying slow-cycling BrdU/PCNA-double positive (BrdU+PCNA+) cells (*Kurki et al., 1986*; *Bravo and Macdonald-Bravo, 1987*). At day 0, the majority of PCNA+ cells are also positive for BrdU, demonstrating the efficiency of our one-week pulse labeling in detecting most cell proliferation (*Figure 3A*). The overall amount of BrdU+PCNA+ cells then drops dramatically by day 28, and only clusters of cells remain throughout the dental epithelium in both types of teeth. Additionally, post-mitotic, single BrdU+ cells are present in some dental parts, including along the DL and in the more superficial regions of the OE. The latter cells likely represent the quick turnover of the epithelium itself. At day 58, the few remaining BrdU+PCNA+ cells representing putative slow-cycling stem/progenitor cells are exclusively concentrated in two distinct regions of pleurodont teeth, the OE and the deep part of the DL (DDL). Interestingly, however, double positive cells are more scarce in the DL of acrodont teeth while remaining in the OE (*Figure 3A*). Our quantification of BrdU+PCNA+ cells in the pleurodont and acrodont epithelial regions confirms these findings, although the total number of BrdU+PCNA+ cells is relatively similar between the two dentitions (d58; *Figure 3C*). Particularly, inspection of the distribution of BrdU+PCNA+ cells in four different epithelial subdivisions, from the OE to the SDL, highlights a high proliferative DDL in pleurodont teeth at day 0 (region III; *Figure 3B,D*). Since the DDL represents the SDL budding site, we presume that this proliferation pattern may trigger, at least in part, the observed directional growth of the pleurodont SDL towards the lingual portion of the jaw. Consistent with this hypothesis, acrodont teeth show a relatively reduced proportion of BrdU+PCNA+ cells in the DDL region, likely reflecting the lack of directional growth of non-regenerating SDL. At day 58, comparisons of the proportion of BrdU+PCNA+ cells confirm our previously observed variation in the location of LRCs in acrodont and

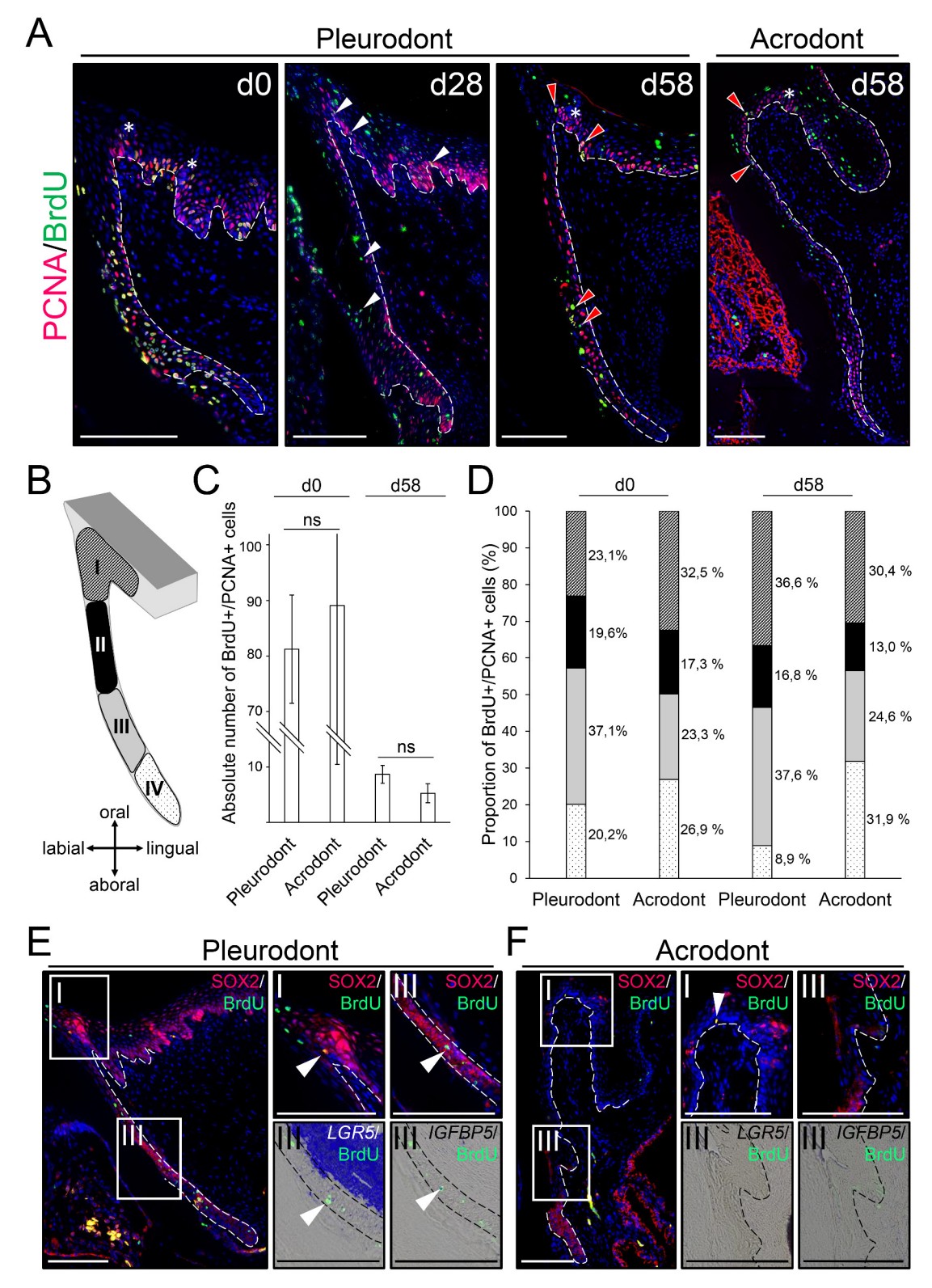

**Figure 3.** Slow-cycling label-retaining cells (LRCs) accumulate in two different regions of bearded dragon teeth. (A) Double IHC for PCNA (red staining) and BrdU (green) in pleurodont (left and middle panels) or acrodont (right) tooth sections collected from bearded dragons at 0 (d0), 28 (d28), and 58 (d58) days of chase after the first week of BrdU pulse. White arrowheads indicate the scattering of BrdU/PCNA-double positive cells throughout the dental epithelium at d28, and red arrowheads show the clustering of BrdU/PCNA-double positive LRCs in both OE and DL at d58. White asterisks

*Figure 3 continued on next page*

*Figure 3 continued*
indicate the position of taste buds in the OE, and white dashed lines delimitate the epithelium-mesenchyme junction. Scale bars: 100 μm. (B) Schematic drawing of the dental subdivisions used for cell quantification: I, OE, defined as a portion of epithelium directly lingual to the DL; II, external DL; III, deep DL; IV, SDL. (C, D) Total amount of BrdU/PCNA-double positive cells in the entire dental epithelium (C; dental zones I-IV) or relative distribution of BrdU/PCNA-double positive cells in the individual dental subdivisions (D; color-code as in (B)) of pleurodont and acrodont teeth at d0 and d58, *n* = 3 per group (ns, non-significant). (E, F) IHC for SOX2 (red staining) and BrdU (green; left, middle top, and right top panels) or double IHC/ISH for *LGR5* (middle bottom) or *IGFBP5* (right bottom) with BrdU in parallel sections of pleurodont (E) and acrodont (F) teeth at d58. Insets show high magnifications of OE (dental zone I as in (C)) or DDL (dental zone III) positive stainings. White arrowheads indicate co-expression of SOX2, *LGR5*, or *IGFBP5* with BrdU. White or black dashed lines delimitate the epithelium-mesenchyme junction. Scale bars: 100 μm.
DOI: https://doi.org/10.7554/eLife.47702.005
The following source data is available for figure 3:

**Source data 1.** Quantification of putative stem/progenitor cells in dental tissues.
DOI: https://doi.org/10.7554/eLife.47702.006

pleurodont teeth. In particular, the DDL and OE regions represent together about three quarters of all LRCs in pleurodont teeth, thus supporting the existence of two distinct sites of putative stem/progenitor cells (*Figure 3D*). In contrast, LRCs are more equally distributed between the DDL, OE, and SDL in acrodont teeth, indicating that altered distribution of LRCs is associated with lack of cyclic tooth replacement (*Figure 3C,D*). To further assess the potential stemness of identified LRCs in bearded dragon dentition, we analyzed the expression of conserved putative stem/progenitor cell markers previously shown to be associated with dental tissues in vertebrates. Of note, tooth replacement is yet poorly studied in polyphyodont species, and only one dental progenitor marker, SOX2, has been validated in several vertebrate species (*Juuri et al., 2012*; *Juuri et al., 2013*; *Martin et al., 2016*). However, hair follicle stem cell markers such as *LGR5* and *IGFBP5* have also been shown to co-localize with LRCs at least in the leopard gecko DL (*Handrigan et al., 2010*). Therefore, we compared the expression pattern of these three markers in pleurodont versus acrodont teeth of our bearded dragon model, using SOX2 IHC or *LGR5*/*IGFBP5* ISH together with BrdU detection (*Figure 3E,F*). Consistent with the BrdU-retaining pattern in pleurodont teeth, we first confirmed the co-expression of SOX2/*LGR5*/*IGFBP5* or only SOX2 within the DDL and OE LRC sites, respectively (*Figure 3E*). However, whereas acrodont teeth show similar co-expression of SOX2 in the OE LRCs, no *LGR5*, *IGFBP5*, or SOX2 expression could be co-detected in the scattered BrdU+ cells of acrodont DL. Altogether, these data clearly indicate the presence of separate putative dental epithelial stem/progenitor populations in bearded dragon, including one pleurodont-specific DDL population relatively similar to the leopard gecko DL (*Handrigan et al., 2010*), and one shared pleurodont/acrodont OE population directly adjacent to the tooth and resembling the superficial taste/tooth junction recently identified in the spotted catshark (*Martin et al., 2016*).

## Cell migration in bearded dragon dentition

The striking similarities between the SOX2+ stem/progenitor populations in the superficial OE of bearded dragon and spotted catshark suggest that cell migration from the OE could also contribute to tooth replacement in reptiles, as shown in sharks (*Martin et al., 2016*). To test this hypothesis, we performed cell tracking experiments in bearded dragons both in vivo and ex vivo, using the lipophilic dye DiI (*Figure 4A,B*). Labeling was carried out in the OE directly adjacent to pleurodont or acrodont teeth, and samples were collected at regular time intervals after DiI administration. Strikingly, in vivo tracking in pleurodont dentition revealed that within two weeks, cells labeled in the OE (d0) shifted to the DL (d7) and then SDL (d14), but also contributed to replacement teeth (d14; *Figure 4A*). A similar ending-point was observed after four weeks of tracing (data not shown), and a corresponding pattern was observed in acrodont teeth, indicating that migration from the OE still occurs despite the lack of replacement. Importantly, imaging of single ex vivo dental cultures confirmed the observed migratory pathway of DiI labeled cells along the DL, from OE to SDL (*Figure 4B*), further demonstrating the contribution of OE progenitor/stem cells to tooth replacement. Particularly, the similar migration pattern observed in both dentition types suggests that, while not sufficient to induce replacement, the maintained growth of the DL/SDL might rely on this OE population. To test this hypothesis, we assessed the functional role of the putative OE stem/progenitor zone by manually removing the entire proximal OE in dental tissue cultures (*Figure 4C*). As

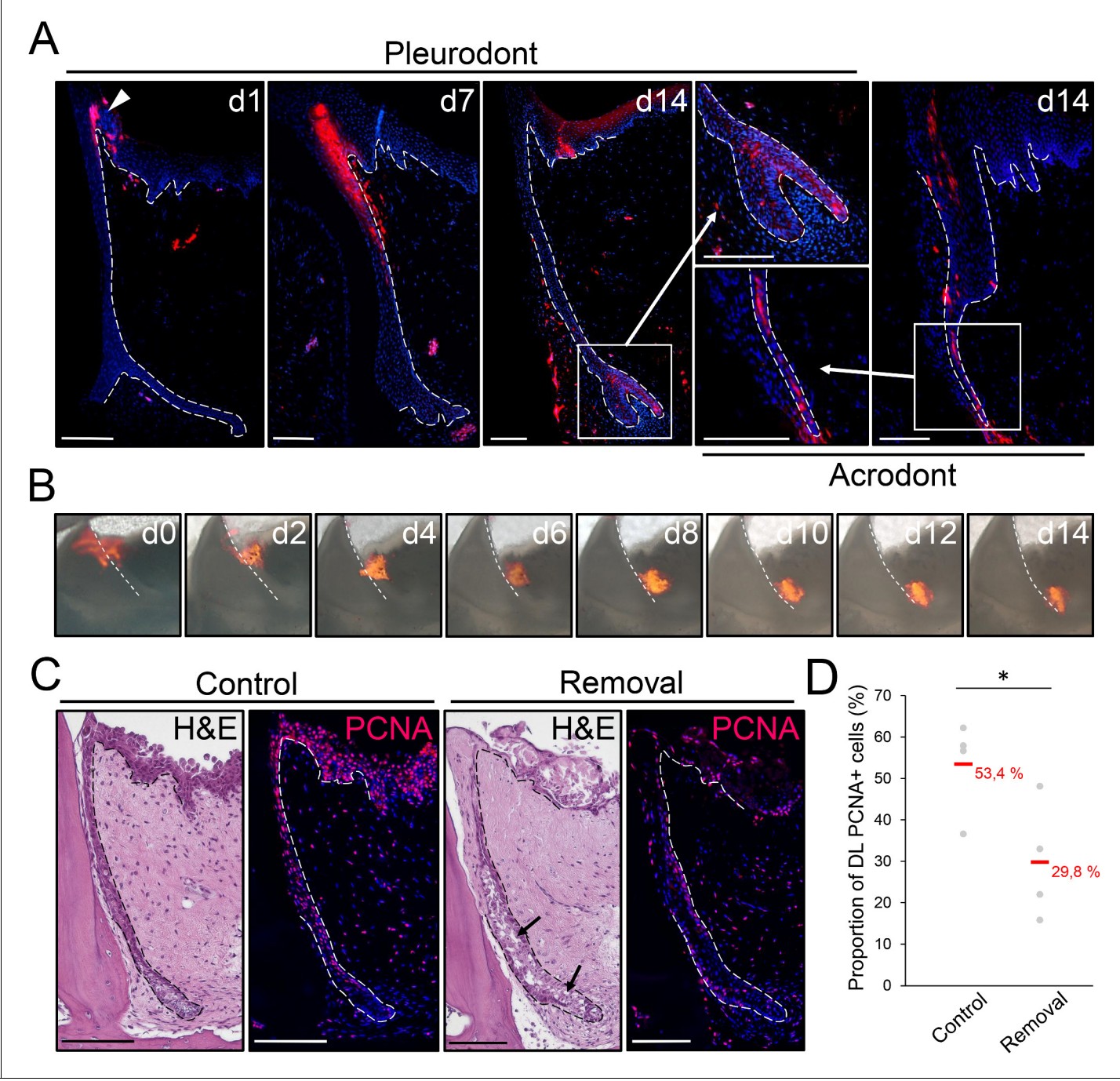

**Figure 4.** The OE contributes to SDL both in vivo and ex vivo. (**A**) DiI fluorescence (red staining) in pleurodont (left and middle panels) or acrodont (right) tooth sections collected from bearded dragons at 1 (**d1**), 7 (**d7**), and 14 (**d14**) days after in vivo DiI administration at the OE-DL junction. Cell nuclei are counterstained with DAPI (blue). The arrowhead at d0 indicates the site of administration of DiI in the OE. High magnifications of the SDL in pleurodont and acrodont teeth are shown at d14 (insets). White dashed lines delimitate the epithelium-mesenchyme junction. Scale bars: 100 μm. (**B**) DiI fluorescence (red) in cultured pleurodont dental tissue slices, imaged every other day (**d0–d14**) over a two-week period following initial dye administration (**d0**). White dashed lines indicate the border between the erupted tooth and the DL. Scale bars: 500 μm. (**C**) H and E and PCNA IHC (PCNA; red staining) in paraffin sections from cultured tissue slices of pleurodont teeth. The OE region near the OE-DL junction of teeth was removed on one side of the jaw (removal), and the opposing, equivalent teeth were used as controls (control). Black arrows indicate disrupted DL and SDL tissues in the removal experiment after one week of culture. Dashed lines separate epithelium from mesenchyme. Scale bars: 100 μm. (**D**) Quantification of the proportion of PCNA-positive cells in both DL and SDL in one-week dental tissue cultures with intact (control) or removed (removal) OE, as described in (**C**). Red values indicate mean values, n = 4 per group (*, p-value<0.05).

*Figure 4 continued on next page*

*Figure 4 continued*

DOI: https://doi.org/10.7554/eLife.47702.007

The following source data is available for figure 4:

**Source data 1.** Quantification of proliferating cells in dental tissues with intact or removed OE.

DOI: https://doi.org/10.7554/eLife.47702.008

shown in *Figure 4C*, disruption of the OE leads to a significant decrease in proliferating DL/SDL cells (see also quantification in *Figure 4D*) as well as to some degeneration of DL/SDL structures in some samples, when compared to intact cultures, thus confirming the regulatory role of bearded dragon OE on DL/SDL growth and, eventually, tooth replacement.

## Differential gene expression patterns in pleurodont and acrodont teeth

To obtain the first high-throughput molecular profiling of tooth replacement in vertebrates, we exploited the unique heterodont features of the bearded dragon to analyze and directly compare the transcriptome of pleurodont versus acrodont dental tissues. We particularly focused on the SDL, which shows divergent expression patterns in our candidate-based approach (see *Figure 2*), and the dental mesenchyme surrounding it, using staining-guided laser microdissection (see microdissected areas in *Figure 5A*) and Illumina RNA sequencing. At a cut-off of False Discovery Rate (FDR)-adjusted p-values<0.05, we obtained 206 and 368 differentially expressed genes in the SDL and dental mesenchyme, respectively (*Figure 5A*, *Supplementary File 1*). Importantly, candidate genes already identified in the bearded dragon SDL, including *PITX2*, *LEF1*, and *SOX2* were all retrieved in our transcriptome data, thus supporting our ISH experiments (see *Figure 2B*). Furthermore, apart from *LEF1* that was only slightly overexpressed in the pleurodont SDL (pleurodont/acrodont fold change of 2.2), many homeotic transcriptional regulators (*BARX1*, *ALX1/ALX4*, *SIX3*, *DLX2*) and members of classical signaling pathways (*BMP7*, *WNT11*, *FGF7*) were highly differentially expressed between the two dentition types. A more thorough analysis of gene ontology (GO) category distribution among all differentially regulated genes indicates that more than 45% belong to the category 'developmental process' (*Figure 5B*). Surprisingly, however, our list of developmental genes revealed only a partial overlap (18%) with genes already known to be expressed in vertebrate dental tissues (*Figure 5B*), indicating that regulation of the SDL in polyphyodont dentitions involves many genes and signaling pathways still uncharacterized and/or non-expressed in the monophyodont mouse dentition. The differential expression of a selection of newly identified genes was confirmed using quantitative PCR (*Figure 5C*). Of the selected genes, *SIX3* and *ISL1* were significantly upregulated in the pleurodont epithelium, *FOXI1* and *BARX1* were downregulated in the pleurodont mesenchyme, and *ALX1* was significantly enriched in both pleurodont epithelial and mesenchymal tissues (*Figure 5A,C*). To further characterize their expression pattern during odontogenesis, ISH analysis of *ALX1*, *SIX3* and *ISL1* was performed at both early postnatal (stage equivalent to transcriptomic analysis) and embryonic developmental stages (*Figure 6*). Interestingly, the relatively higher expression of *ALX1* and *SIX3* in pleurodont versus acrodont tissues was apparent in all stages tested by ISH, thus indicating variations in gene expression starting from early tooth developmental stages. However, similarly to the different classical dental markers tested (see *Figure 2*), differential *ISL1* expression between the dentition types was only apparent at late tooth developmental stages, suggesting that this gene plays a role in tooth replacement. As already revealed by our transcriptomic data, the expression pattern of *ISL1* and *SIX3* was largely restricted to the SDL postnatally in both types of teeth, while being detected in both epithelium and dental papilla mesenchyme at embryonic dental stages. In contrast, *ALX1* was confined to the developing pleurodont mesenchymes at all embryonic stages, but was later detected both in the pleurodont SDL and surrounding mesenchyme of postnatal teeth, thus supporting our RNA sequencing results and indicating dynamic expression pattern during tooth development and SDL patterning.

## Discussion

The lack of lifelong replacement of oral dentition in many different lineages including classical model organisms has hampered the analysis of tooth regeneration in vertebrates. Furthermore, studies

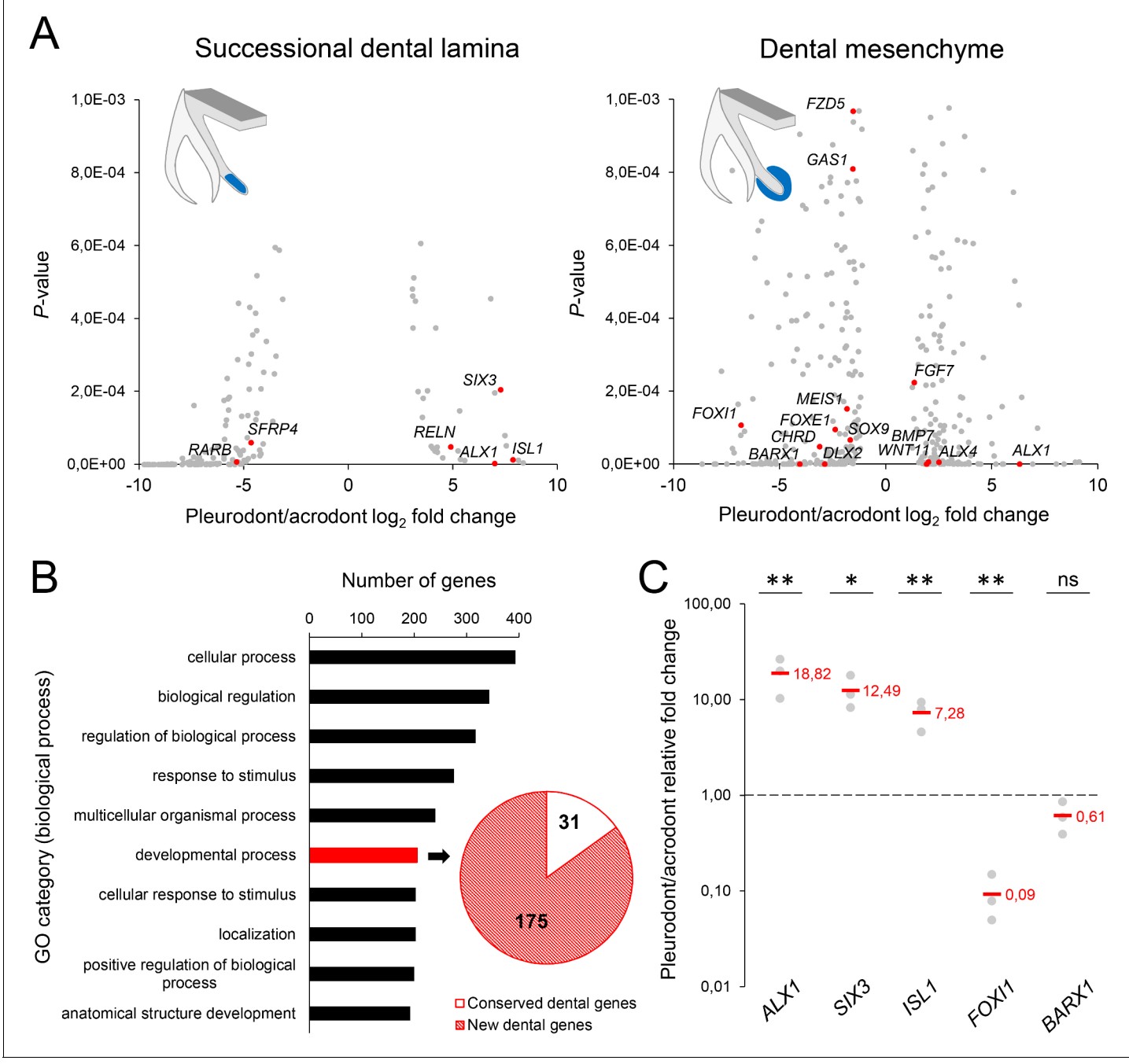

**Figure 5.** RNA sequencing of micro-dissected SDL and dental mesenchyme tissues reveals putative new genes involved in tooth replacement. (**A**) Volcano plots of significantly differentially expressed genes (FDR-corrected p-values<0.05) between pleurodont and acrodont SDL (*n* = 206, left plot) and dental mesenchyme (*n* = 368, right) tissues at early postnatal stage (four dph), *n* = 3 per group. The x- and y-axes display the log₂ fold change (pleurodont with respect to acrodont) and associated *P*-values, respectively. Positive log₂ values indicate gene upregulation in pleurodont teeth, and red plot points highlight some key developmental genes and transcription factors (see main text). Schematic drawings illustrate the laser-microdissected areas (blue color) used for transcriptomics in each dental tissue type. (**B**) Gene ontology (GO) analysis of all differentially expressed genes between pleurodont and acrodont dental tissues (*n* = 574), as identified in (**A**). The ten top-ranking biological process categories by gene number are shown. The pie chart depicts the gene distribution from the category 'developmental process'. Genes already known to be associated with vertebrate tooth development were defined as 'conserved dental genes' (white portion; *n* = 31), whereas other genes were categorized as 'new dental genes' (red shaded portion; *n* = 175). (**C**) Quantitative PCR of *ALX1*, *SIX3*, *ISL1*, *FOXI1*, and *BARX1* in dental tissues from pleurodont and acrodont teeth. The x- and y-axes display the gene names and relative fold change (pleurodont with respect to acrodont), respectively. The dashed line depicts

*Figure 5 continued on next page*

*Figure 5 continued*

the position of a fold change of 1.00 (equal expression). Red values indicate mean values, *n* = 3 per gene (ns, non-significant; *, p-value<0.05; **, p-value<0.01).

DOI: https://doi.org/10.7554/eLife.47702.009

The following source data is available for figure 5:

**Source data 1.** Quantitative PCR of newly identified dental genes in dental tissues.

DOI: https://doi.org/10.7554/eLife.47702.010

investigating polyphyodont species have so far largely focused on single- or cross-species comparisons of selected candidate genes based on mouse expression patterns, and the exact molecular basis of tooth renewal is still unclear. Here, we show that the bearded dragon, one of the rare vertebrate species with both polyphyodont and monophyodont teeth, constitutes a key model for filling this gap, allowing direct comparison of different dentition types. As already shown in non-teleost polyphyodont species, our findings first confirm the critical importance of both DL and SDL structures in tooth renewal. However, in contrast to a previous hypothesis (*Buchtová et al., 2012*; *Dosedělová et al., 2015*), the early degradation of the SDL is not the only mechanism restricting vertebrate tooth replacement. Indeed, bearded dragons maintain a proliferative epithelial SDL structure even in non-replacing teeth, indicating that the SDL maintenance is independent of and not strictly associated with tooth renewal. These data also contrast with previous observations made in acrodont teeth from chameleons, the second family of acrodont lizards, where the SDL was shown to diminish in size at postnatal stages independently of increased apoptosis (*Buchtová et al., 2012*). The bearded dragon posterior dentition might thus represent an intermediary stage between polyphyodonty and complete monophyodonty, at least at the cellular level. In favor of that, and in contrast to chameleons, the implantation mode in most agamids is similar to intermediate pleurodont-acrodont dentition patterns observed in extinct acrodont iguanian lizards (*Averianov et al., 1996*; *Simões et al., 2015*). Haridy even recently proposed that the posterior tooth phenotype in bearded dragons might result from pleurodont to acrodont remodeling changes through ontogeny (*Haridy, 2018*). The latter hypothesis is coherent with our developmental data, as no clear morphological variations were observed in the two types of developing teeth, except for the length of the embryonic primary DL. The origin of such difference in the developing DL is unclear but might be linked to the delayed initiation of pleurodont teeth, when compared to acrodont teeth, which would extend the timing of DL proliferation. Importantly, differences were also noticed in the directional growth of SDL at hatchling and postnatal stages, with the pleurodont and acrodont SDL growing lingually or projecting more downwards towards the jaw bone, respectively. This differential directional SDL growth likely constitutes a key process to accommodate the replacement tooth, and was accompanied by an asymmetric expression pattern of conserved dental genes such as *LEF1* and *PITX2*, but also of proliferation, towards the pleurodont SDL tip. Interestingly, focal localization of *LEF1* expression and/or Wnt signaling activity was already noticed in the squamate SDL (*Handrigan et al., 2010*; *Gaete and Tucker, 2013*), alligator DL bulge (*Wu et al., 2013*), shark SDL (*Rasch et al., 2016*), and ferret SDL (*Jussila et al., 2014*). However, our large-scale transcriptomic analysis now indicates that a large number of yet uncharacterized developmental genes and signaling pathways are, in fact, likely associated with SDL outgrowth and/or induction. Furthermore, besides differences in gene expression dynamics and regionalization between pleurodont and acrodont dentitions, our data show drastic changes in gene expression levels in both SDL and surrounding mesenchymal tissues, indicating the importance of multi-level dental organization for initiation of tooth replacement. Along this line, the overall conserved expression levels of Wnt signaling members in our comparative pleurodont-acrodont transcriptome analysis suggest that this conserved pathway might primarily regulate SDL growth, a process maintained in bearded dragon acrodont dentition, rather than directly induce tooth replacement. This hypothesis is consistent with our observations that separate processes regulate SDL growth and initiation of replacement teeth, further indicating that SDL maintenance is not causatively associated with tooth replacement. Among newly identified developmental factors differentially regulated between monophyodont and polyphyodont dentitions, several genes encoding homeobox transcription factors were highly upregulated in regenerating teeth. For example, *ALX1*, a regulator of cartilage and craniofacial development in

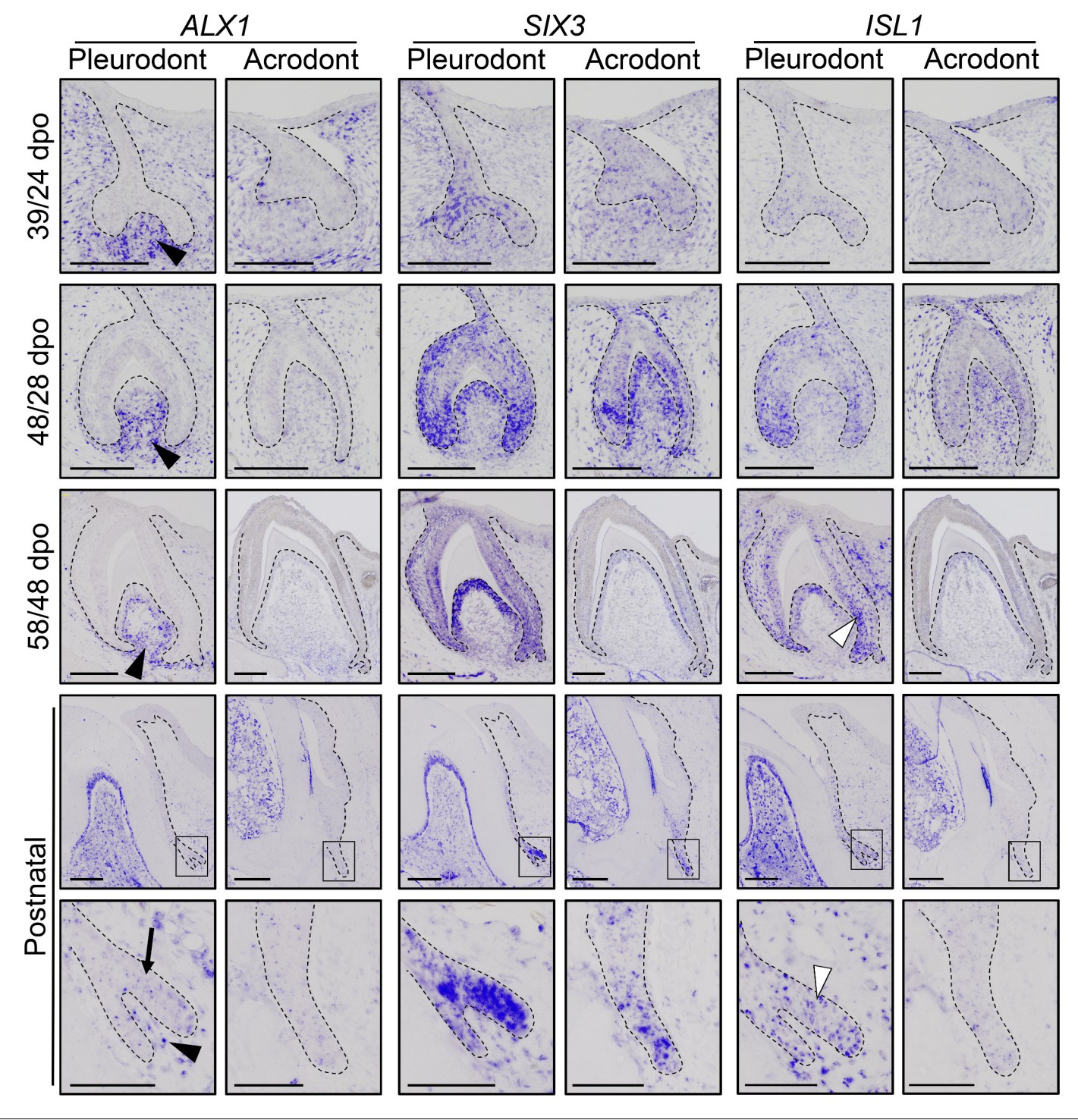

**Figure 6.** Comparative expression pattern of new dental genes in developing bearded dragon dentition. ISH showing the expression of *ALX1* (left panels), *SIX3* (middle), and *ISL1* (right) at various indicated developmental stages in embryonic and early postnatal pleurodont and acrodont dental tissues. As pleurodont teeth are developmentally delayed compared to acrodont teeth (see *Figure 1E–P*), different embryonic stages were used to attain comparable dental developmental stages in the two dentition types (e.g., 39/24 dpo indicates 39 dpo for pleurodont teeth and 24 dpo for acrodont teeth). The positive signal is false-colored to enhance visibility. High magnifications of the SDL region, as indicated by black outlines, are shown for postnatal teeth (bottom panels). Black arrowheads show relatively higher *ALX1* expression in the mesenchymes of embryonic and postnatal pleurodont teeth. In postnatal SDL, *ALX1* transcripts are also present in the epithelial compartment (black arrow). Increased *ISL1* expression in

*Figure 6 continued on next page*

*Figure 6 continued*
pleurodont epithelium is only detected at late developmental stages (white arrowheads). Black dashed lines separate the epithelium from mesenchyme.
Scale bars: 50 µm (bottom panels); 100 µm (other images).
DOI: https://doi.org/10.7554/eLife.47702.011

vertebrates (*Zhao et al., 1993*; *Zhao et al., 1996*; *Dee et al., 2013*), was one of the most highly upregulated genes in both pleurodont SDL and mesenchymal tissues, and it is likely that the severe and relatively early craniofacial phenotypes of *ALX1* mutations (frontonasal dysplasia) before initiation of odontogenesis have hampered identification of this gene as a regulator of tooth development and/or replacement in mammals. Similarly, *SIX3* plays crucial roles in the development of the vertebrate sensory system (*Oliver et al., 1995*; *Loosli et al., 1999*; *Del Bene et al., 2004*), but this gene has not yet been detected in the developing mouse dentition (*Nonomura et al., 2010*). Finally, *ISL1* has been reported to be selectively expressed in the dental epithelium of mouse incisor, while being absent from molar epithelium (*Mitsiadis et al., 2003*), but this tooth-specific patterning has never been linked with continuous tooth growth or replacement in previous vertebrate studies. Based on the dynamic expression pattern observed in the developing epithelial and/or mesenchymal tissues, further investigation of these genes should provide key information on the signaling pathways controlling odontogenesis, including SDL outgrowth and/or initiation of replacement teeth.

As already suggested by several studies in polyphyodont species (*Handrigan et al., 2010*; *Wu et al., 2013*; *Martin et al., 2016*; *Thiery et al., 2017*), our identification of bearded dragon slow-cycling LRCs co-expressing stem/progenitor markers provides further evidence of the importance of epithelial stem/progenitor cell populations in regulating tooth renewal in vertebrates (*Figure 7*). However, the mechanism of continuous tooth replacement identified in our model differs from the leopard gecko, the first inspected lizard species (*Handrigan et al., 2010*), by the presence of two separate sources of putative stem/progenitor cells in the DL and OE (*Figure 7*), thus indicating alternative regenerative strategies and some specialization of stem/progenitor niches in squamates. Particularly, we show that migration from SOX2+ LRCs at the DL/OE junction contributes to the proliferative growth of DL and SDL structures, a process shared by both acrodont and pleurodont teeth (*Figure 7*). These findings support our hypothesis about the intermediate phenotype of bearded dragon acrodont dentition and closely parallel the regenerative dental strategy of sharks, which exhibit a similar LRC population at the taste/tooth junction (*Figure 7*). It is then tempting to speculate that SOX2+ OE cells of bearded dragons could also be bipotent, contributing to both taste and dental lineages (*Martin et al., 2016*). In support of this, SOX2 has been recently identified as a conserved marker of dental progenitors but also of taste bud and non-gustatory epithelial fate in vertebrates (*Ohmoto et al., 2017*). Furthermore, similarly to sharks, the OE of bearded dragons contains a heterogeneous population of cells, consisting of self-renewing epithelial progenitors and taste buds in close proximity to the DL (*Figure 3A* and data not shown). However, anatomical studies have revealed strong heterogeneity in the presence and location of taste buds in lizards and snakes (*Uchida, 1980*; *Schwenk, 1985*), indicating that this potential bipotency may not be a general mechanism conserved across squamates. It is also still unclear if the newly identified OE LRC population exists and contributes to tooth replacement in other reptiles, as the OE has not been analyzed in previous lizard and snake studies despite the strong SOX2 expression observed in this region (*Juuri et al., 2013*; *Gaete and Tucker, 2013*). The presence of one OE LRC population might also be directly linked to a specific tooth replacement rate and/or strategy in squamates, in a similar way as the heterogeneity in DL reported in teleosts with different replacement states (*Fraser et al., 2006*; *Jernvall and Thesleff, 2012*). Indeed, replacement of pleurodont teeth in agamid lizards is relatively slow (*Cooper et al., 1970*) and this study) and based on a 'one-for-one' strategy (a single replacement tooth is formed at any one time for a single tooth position), whereas many squamates exhibit a 'many-for-one' replacement state with several replacement teeth formed ahead of function at a single tooth position. In this context, the conservation of only one putative stem/progenitor cell population in the DL might constitute a more efficient positioning strategy for generating multiple tooth generations, as exemplified by the leopard gecko (*Handrigan et al., 2010*). Our characterization of the second source of LRCs in the bearded dragon pleurodont DL indicates striking similarities with the leopard gecko in terms of location, directly beside the lingually protruding SDL in the DDL,

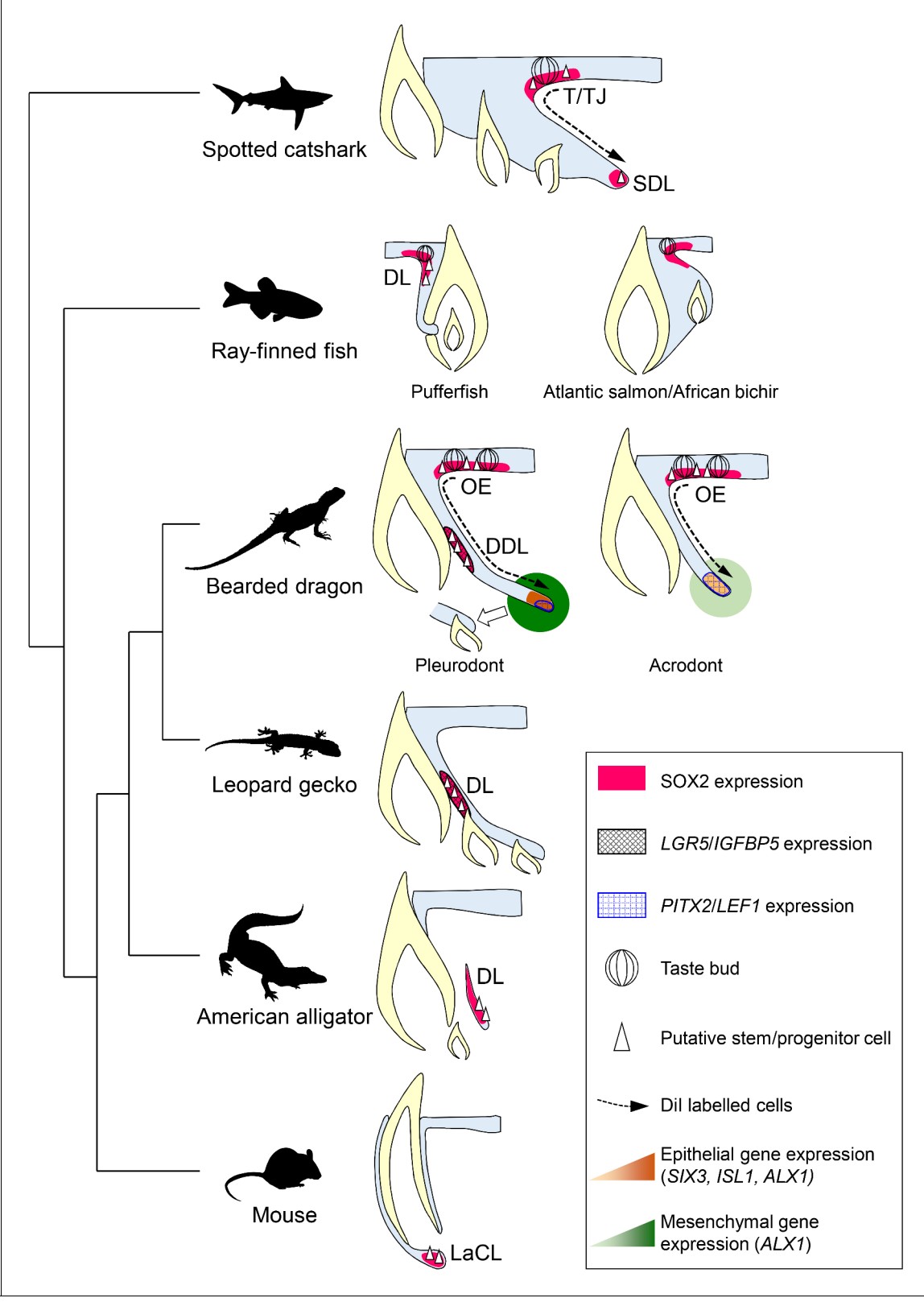

**Figure 7.** Comparative models of cellular and molecular processes controlling oral tooth replacement in polyphyodont species. Schematic drawings of oral tooth replacement strategies in polyphyodont species with molecularly-characterized LRC populations: spotted catshark (*Martin et al., 2016*), pufferfish (*Thiery et al., 2017*), Atlantic salmon (*Vandenplas et al., 2016*), African bichir (*Vandenplas et al., 2016*), bearded dragon pleurodont and acrodont teeth (this study), leopard gecko (*Handrigan et al., 2010*), and American alligator (*Wu et al., 2013*). The putative stem cell niche of

*Figure 7 continued*

monophyodont mouse incisor (*Harada et al., 1999*) is also shown for comparison. In spotted catshark, SOX2-positive putative dental progenitors migrate from the superficial taste/tooth junction (T/TJ) towards the successional dental lamina (SDL). In pufferfish but also cichlid fish (*Fraser et al., 2013*), a SOX2-positive putative dental progenitor niche resides in the most superficial dental lamina (DL). In Atlantic salmon and African bichir, no epithelial LRCs have been identified despite positive SOX2 expression in the OE and outer dental epithelium transition zone. In the leopard gecko, LRCs expressing adult stem cell markers such as *IGFBP5* and *LGR5* reside on the lingual side of the DL. In the American alligator, putative stem cells localize to the distal enlarged bulge of the DL. In both leopard gecko and American alligator, SOX2 expression has been shown to overlap with the epithelial region containing putative stem cells (*Juuri et al., 2013*), but no co-localization studies are available. In mouse incisors, SOX2-positive putative stem cells responsible for continual growth are located in the labial cervical loop (LaCL; *Juuri et al., 2012*). In bearded dragon pleurodont teeth, LRCs are located both in the SOX2-positive oral epithelium (OE; region similar to the T/TJ) and *IGFBP5*/*LGR5*/SOX2-positive DDL (region similar to the gecko DL). During regeneration, cells migrate from the superficial OE towards the SDL, the SDL shows focal expression of SDL marker genes (*PITX2*/*LEF1*), and both the SDL and surrounding mesenchyme exhibit relatively high expression of newly identified dental genes (*ALX1*/*SIX3*/*ISL1*), thus leading to the initiation of replacement tooth. In acrodont teeth, LRCs are also evident in the SOX2-positive OE and cell migration occurs from the superficial OE towards the SDL, thus contributing to SDL growth. However, the acrodont SDL shows scattered expression of SDL markers and low expression of newly identified dental genes, most likely as a result of absence of DL stem/progenitor cells and SDL organization, and no replacement teeth are formed.

DOI: https://doi.org/10.7554/eLife.47702.012

but also in terms of stem/progenitor marker expression profile, suggesting evolutionary conservation of this cell population at least among lizards (*Figure 7*). However, the exact function of putative stem/progenitor niches has remained unclear since their discovery in the leopard gecko. Our pleurodont-acrodont comparisons in bearded dragons now provide the first evidence of functional specialization of putative dental stem/progenitor populations in vertebrates. Specifically, we show that the DL population is likely strictly necessary for tooth replacement, as it is not detected in monophyodont bearded dragon teeth. The altered proliferation pattern, including lack of asymmetry and increased presence of BrdU-retaining cells, as well as differences in gene expression levels and location in the acrodont SDL suggest a key role for the DL population in the complex SDL organization required for initiation of tooth replacement (*Figure 7*). In contrast, the maintained OE population in the whole bearded dragon dentition contributes to SDL maintenance and growth, a process that is not causatively associated with initiation of tooth replacement.

In conclusion, our results demonstrate that the various key dental features of the emerging bearded dragon model offer a powerful system to elucidate the developmental and genetic basis of evolutionary novelty and tooth regeneration in vertebrates. Particularly, the direct comparative analysis of pleurodont and acrodont dentitions indicates the importance of SDL patterning, in addition to maintenance, for vertebrate tooth replacement. This patterning process happens at various levels, including directional growth but also gene expression levels, dynamics, and regionalization, and involves a large number of yet uncharacterized developmental genes and signaling pathways. Finally, the two separate putative stem/progenitor populations identified in bearded dragon pleurodont teeth reveal a new vertebrate dental regenerative strategy associated with cell migration and functional specialization of putative stem/progenitor cells.

# Materials and methods

## Key resources table

| Reagent type | Designation | Source or reference | Identifiers | Additional information |
|---|---|---|---|---|
| Antibody | PCNA (mouse monoclonal) | BioLegend | cat# 307901, RRID:AB_314691 | 1:300 |
| Antibody | BrdU (rat monoclonal) | Abcam | cat# ab6326, RRID:AB_314691 | 1:200 |
| Antibody | SOX2 (rabbit polyclonal) | Abcam | cat# ab97959, RRID:AB_2341193 | 1:200 |

*Continued on next page*

*Continued*

| Reagent type | Designation | Source or reference | Identifiers | Additional information |
|---|---|---|---|---|
| Antibody | DIG conjugated to alkaline phosphatase (sheep polyclonal) | Sigma-Aldrich | cat# 11093274910, RRID:AB_2734716 | 1:2000 |
| Antibody | Alexa Fluor-488 (goat anti-rat IgG) | Thermo Fisher Scientific | cat# A-11006, RRID:AB_2534074 | 1:400 |
| Antibody | Alexa Fluor-568 (goat anti-rabbit IgG) | Thermo Fisher Scientific | cat# A-11011, RRID:AB_143157 | 1:400 (with SOX2), 1:600 (with PCNA) |
| Sequence-based reagent | PITX2 | This paper | PCR primers | Forward primer (fp) ATGAACTGCCTGAAAGACGC; reverse primer (rp) CATCAGGCCGTTGAATTGGG |
| Sequence-based reagent | LEF1 | This paper | PCR primers | (fp) GCCACCGACGAGATGATCC; (rp) GTGCGAAGGATGTGTCCCTG |
| Sequence-based reagent | SHH | *Di-Poï and Milinkovitch (2016)* | PCR primers | (fp) CAAGCAGTTCCATCCCCAAC; (rp) GCCCAGCTATGCTCCTCAAT |
| Sequence-based reagent | NOTCH1 | This paper | PCR primers | (fp) GCCACATC CTGGACTACGAC; (rp) GGAATGTCCAGGTTCCCGAG |
| Sequence-based reagent | LGR5 | This paper | PCR primers | (fp) GGAATGTCCAGGTTCCCGAG; (rp) GGCACTAGTGAATTGCTGGGG |
| Sequence-based reagent | IGFBP5 | This paper | PCR primers | (fp) GCAGAGGAGACCTTCCAACC; (rp) CTGAGGGCTTCTCACACCAG |
| Sequence-based reagent | ALX1 | This paper | PCR primers | (fp) CTGTCTCCCGTGAAAGGC AT; (rp) TAACAG AAGTGGGTGACTGCC |
| Sequence-based reagent | SIX3 | This paper | PCR primers | (fp) TGCCCACG CTCAACTTTTC; (rp) CCGCCGAA CTGTGAGTAGGA |
| Sequence-based reagent | ISL1 | This paper | PCR primers | (fp) ACCTGCTTT GTTAGGGACGG; (rp) CGTCGTG TCTCTCCGGAC TA |
| Sequence-based reagent | ALX1 | This paper | qPCR primers | (fp) GCAGTTCC GTTGTGACTTC; (rp) ATCTGTCC GAGGTGAATGG |
| Sequence-based reagent | SIX3 | This paper | qPCR primers | (fp) CTCTACCAC ATCCTGGAGAAC; (rp) TTCCTGG TAGTGAGCTTCG |
| Sequence-based reagent | ISL1 | This paper | qPCR primers | (fp) TGCGGCA ATCAAATCCAC; (rp) GGTTACATT CCGCACACTTC |
| Sequence-based reagent | FOXI1 | This paper | qPCR primers | (fp) GGCTATAC TGGTTCAGTCCTC; (rp) ACTTCA GTGCCCTCTCTTG |
| Sequence-based reagent | BARX1 | This paper | qPCR primers | (fp) AAGGTGGA GGGCTTGAATC; (rp) TGTCAACT GCTCGCTACTG |
| Sequence-based reagent | ACTB | This paper | qPCR primers | (fp) CCTGGAGA AGAGCTACGAAC; (rp) AGAAAG ACGGCTGGAAGAG |
| Commercial assay or kit | RNeasy Plus Micro Kit | Qiagen | cat#/ID: 74034 | |
| Commercial assay or kit | Ovation SoLo RNA-Seq Library Preparation Kit | Nugen | cat#/ID: 0501–32 | |
| Commercial assay or kit | RNA 6000 Pico Kit | Agilent | cat#: 5067–1513 | |
| Commercial assay or kit | QuantiTect Reverse Transcription Kit | Qiagen | cat#/ID: 205311 | |
| Commercial assay or kit | iTaq Universal SYBR Green Supermix | Bio-Rad | cat#: 1725121 | |
| Commercial assay or kit | TUNEL in situ cell death detection | Roche | cat#: 11684795910 | |
| Chemical compound, drug | BrdU | Sigma-Aldrich | cat#: B5002 | Administered at 80 mg/kg |

*Continued on next page*

*Continued*

| Reagent type | Designation | Source or reference | Identifiers | Additional information |
|---|---|---|---|---|
| Chemical compound, drug | DiI | Biotium | cat#: 60010 | Dissolved into ethanol at 25 mg/ml; working solution further diluted in 0.3M sucrose at 5 mg/ml |
| Chemical compound, drug | Fluoroshield mounting medium | Sigma-Aldrich | cat#: F6057 | |
| Chemical compound, drug | Dako Faramount aqueous medium | Agilent | cat#: S302580 | |
| Software, algorithm | Nrecon | Bruker | | |
| Software, algorithm | Advanced 3D Visualization and Volume Modeling (Amira) | Thermo Fisher Scientific | RRID:SCR_007353 | |
| Software, algorithm | Fiji/ImageJ | *Schindelin et al. (2012)* | RRID:SCR_002285 | |
| Software, algorithm | Adobe Photoshop CC | Adobe | RRID:SCR_014199 | |
| Software, algorithm | NIS-Elements | Nikon | RRID:SCR_014329 | |
| Software, algorithm | Leica Application Suite X | Leica | RRID:SCR_013673 | |
| Software, algorithm | Microsoft Excel | Microsoft | RRID:SCR_016137 | |
| Software, algorithm | Zen | Zeiss | RRID:SCR_013672 | |
| Software, algorithm | FastQC | http://www.bioinformatics.babraham.ac.uk/projects/fastqc/ | RRID:SCR_014583 | |
| Software, algorithm | Trimmomatic | *Bolger et al. (2014)* | RRID:SCR_011848 | |
| Software, algorithm | SortMeRNA | *Kopylova et al. (2012)* | RRID:SCR_014402 | |
| Software, algorithm | STAR | *Dobin et al. (2013)* | RRID:SCR_015899 | |
| Software, algorithm | edgeR | *Robinson et al. (2010)* | RRID:SCR_012802 | |
| Software, algorithm | PALM Robo | Zeiss | RRID:SCR_014435 | |
| Software, algorithm | CFX Manager | Bio-Rad | RRID:SCR_017251 | |
| Other | PALM MicroBeam device | Zeiss | | |

## Sample collection

All embryonic and postnatal stages of bearded dragons (*Pogona vitticeps*) were obtained from our animal facility at the University of Helsinki. For embryonic stages, fertilized eggs were incubated on a moistened vermiculite substrate at 29.5°C, and embryos were removed at regular interval after oviposition to obtain stages of interest. Embryos were staged on the basis of their external morphology according to a complete developmental table available for this species (*Ollonen et al., 2018*).

## CT-scanning and 3d rendering

For general tooth morphology, heads were fixed overnight at 4°C in 4% paraformaldehyde (PFA), dehydrated in an increasing alcohol series, and CT-scanned using a Bruker SkyScan 1272 instrument (parameters: 70 kV, 142 µA, Al 0.5 mm filter, 13.2 µm resolution). For visualizing the dental lamina (DL), dissected jaws were fixed as above, dehydrated to 70% ethanol, and stained for 2–4 weeks

with 0.3% phosphotungstic acid (PTA) in 70% ethanol, as described before (*Metscher, 2009*). Stained jaws were then CT-scanned in 70% ethanol (90 kV, 111 µA, Al 0.5 + Cu 0.038 mm filters, 2 µm resolution). All CT-scans were reconstructed using Bruker NRecon 1.7.0.4 software, and 3D volume rendering and segmentation of DL and/or teeth were done manually using Advanced 3D Visualization and Volume Modeling 5.5.0 (RRID:SCR_007353).

## Histology, immunohistochemistry (IHC), and apoptosis detection on paraffin sections

Following dissection, tissues were fixed overnight in 4% PFA at 4°C, and late embryonic stages showing mineralization were decalcified for 1–12 weeks, depending on sample size, in 20% ethylenediaminetetraacetic acid (EDTA) containing 0.5% PFA. Fixed and decalcified tissues were then progressively dehydrated into 100% methanol, embedded into paraffin blocks using a Leica ASP200 embedding machine, and sectioned at 7 µm in sagittal (for pleurodont teeth) or coronal (acrodont teeth) plane with a Microm HM355 microtome. Hematoxylin and eosin (H and E) staining of dental sections was done according to standard protocols, and images were acquired with an AX70 microscope, DP70 camera, and DP controller 1.2.1.108 software (Olympus). IHC fluorescent staining was performed as described before (*Di-Poï and Milinkovitch, 2016*), using overnight incubation at 4°C with primary antibodies known to recognize reptile and/or chicken epitopes: proliferating cell nuclear antigen (PCNA; 1:300, mouse monoclonal, BioLegend, cat# 307901, RRID:AB_314691), 5-bromo-2'-deoxyuridine (BrdU; 1:200, rat monoclonal, Abcam, cat# ab6326, RRID:AB_305426), and SRY-box 2 (SOX2; 1:200, rabbit polyclonal, Abcam, cat# ab97959, RRID:AB_2341193). Last, incubation with Alexa Fluor-conjugated secondary antibodies (Alexa Fluor-488: goat anti-rat IgG, Thermo Fisher Scientific, cat# A-11006, RRID:AB_2354074; Alexa Fluor-568: goat anti-rabbit IgG, Thermo Fisher Scientific, cat# A-11011, RRID:AB_143157) was performed for 1 hr at room temperature (RT), and slides were mounted with Fluoroshield mounting medium (Sigma-Aldrich) containing 4',6'-diamidino-2-phenylindole (DAPI). Nuclear DNA fragmentation of apoptotic cells was labeled in situ using the TUNEL method (in situ cell death detection kit, Roche), according to the manufacturer's instructions. Before TUNEL staining, tissues were pre-treated with proteinase K (10 µg/ml, 15 min at RT) and Triton X-100 (0.2%, 10 min at RT). Positive controls were performed by treating tissues with DNAse I before staining. Images were acquired either with a Nikon Eclipse 90i widefield microscope and Nikon NIS-Elements Advanced Research 4.30.01 software (RRID:SCR_014329), or with a Leica DM5000B widefield microscope and Leica Application Suite X (LAS X) 3.4.2 12.4.18 software (RRID: SCR_013673). Images were processed with Fiji/ImageJ (RRID:SCR_002285; *Schindelin et al., 2012*) and/or Adobe Photoshop CC (RRID:SCR_014199) using levels adjustment.

## In situ hybridization (ISH)

Digoxigenin (DIG)-labeled antisense riboprobes corresponding to *Pogona vitticeps* paired-like homeodomain 2 (*PITX2*, 544 bp), lymphoid enhancer binding factor 1 (*LEF1*, 460 bp), sonic hedgehog (*SHH*, 931 bp), notch 1 (*NOTCH1*, 865 bp), leucine rich repeat containing G protein-coupled receptor 5 (*LGR5*, 754 bp), insulin-like growth factor-binding protein 5 (*IGFBP5*, 820 bp), ALX homeobox 1 (*ALX1*, 722 bp), sine oculis homeobox homolog 3 (*SIX3*, 639 bp), and ISL LIM homeobox 1 (*ISL1*, 675 bp) genes were designed based on the publicly available *Pogona vitticeps* genome sequence (*Georges et al., 2015*). ISH was performed on paraffin sections as described previously (*Eymann et al., 2019*), using a hybridization temperature of 63°C. Following hybridization, sections were washed and blocked from non-specific antibody binding with blocking solution (2% Roche blocking reagent, 5% goat serum) before incubating overnight at 4°C (or 1 hr at RT) with anti-DIG antibodies conjugated to alkaline phosphatase (1:2000, sheep polyclonal, Sigma-Aldrich, cat# 11093274910, RRID:AB_2734716). A staining solution containing 5-bromo-4-chloro-3-indolyl phosphate and nitro blue tetrazolium was applied for 1–3 days at RT to visualize hybridization. Finally, slides were mounted using Dako Faramount aqueous medium (Agilent) and imaged with an Olympus AX70 microscope. Images were processed with Adobe Photoshop CC (RRID:SCR_014199), and ISH staining was enhanced for visualization by fake-coloring the selected blue pixels with the 'Select color range' and 'Brush' tools.

## Pulse-chase BrdU labeling and quantification

5-bromo-2'-deoxyuridine (BrdU, Sigma-Aldrich) was orally administered to *Pogona vitticeps* ($n$ = 10) twice daily (80 mg/kg body weight) for a period of 7 days. Animals were euthanized 0 ($n$ = 4), 28 ($n$ = 3), or 58 ($n$ = 3) days after BrdU pulse labeling. A later time point (112 days after pulse) was also tested, resulting in none or very few quantifiable LRCs. Chase time points were selected based on previous works on dental tissues (*Handrigan et al., 2010*; *Wu et al., 2013*; *Martin et al., 2016*). For assessing cell proliferation, dental tissues were fixed and processed as described above, and PCNA/BrdU-double IHC was performed on sectioned acrodont and pleurodont teeth at each time point. The dental epithelium was divided into four distinct regions (see *Figure 3B*), and BrdU/PCNA-double positive cells were quantified manually in each region using the 'Count tool' on Adobe Photoshop CC (RRID:SCR_014199). For each time point, cell counts were done using a minimum of three animal replicates, with 2–4 acrodont or pleurodont teeth per replicate and four sections per tooth covering similar coronal (acrodont teeth) or sagittal (pleurodont teeth) sectioning planes. Statistical significance was evaluated with Student's *t*-test in Microsoft Excel (RRID:SCR_016137).

## DiI labeling and dental tissue slice culture

DiI (1,1'-dioctadecyl-3,3,3',3'-tetramethylindocarbocyanine perchlorate; Biotium) was first dissolved into 100% ethanol at 25 mg/ml, and subsequently diluted in 0.3M sucrose to achieve a final concentration of 5 mg/ml. Juvenile animals were anesthetized by injection of propofol (1 mg/ml; Sigma-Aldrich) into the ventral coccygeal vein, and small amounts of DiI:sucrose solution were injected into the oral epithelium (OE) using a Hamilton syringe (702 RN, 25 µl volume) and custom needles (small RN, 26 s gauge). Animals were euthanized 1, 7, or 14 days after DiI administration ($n$ = 6), and dental tissue sections were directly mounted with Fluoroshield mounting medium (Sigma-Aldrich). For ex vivo DiI labeling, the skin of dissected jaws from euthanized *Pogona vitticeps* was removed, and about 1–2 mm thick slices were cut through the jaws with a scalpel. DiI:sucrose was administered into the OE using the tip of standard 30G needle syringe. Slices were placed on Corning Transwell-clear 3 µm nucleopore membrane inserts (Sigma-Aldrich) in 6-well plates filled with DMEM/F12 medium (Gibco) containing 10% fetal calf serum, 100 U/ml penicillin-streptomycin (Gibco), 0.1 mg/ml ascorbic acid (Sigma-Aldrich), 1:100 nystatin (Gibco), and 1:100 amphotericin B (Gibco). Slices were cultured at the air-liquid interface (37°C, 5% $CO_2$) for 2 weeks, and imaged every other day with a Lumar V12 microscope and Zen Digital Imaging for Light Microscopy 1.1.2.0 (RRID:SCR_013672). For better visualization, DiI fluorescence images were overlaid with brightfield images. For OE removal experiments, the OE directly adjacent to the tooth in ex vivo tissue slices from one side of the jaw was carefully removed with a needle, without disturbing the mesenchyme or the DL. Slices from corresponding teeth on the other side of the jaw were used as controls ($n$ = 3 animals per group). Slices were cultured for 7 days, as described above, and then fixed overnight at 4°C with 4% PFA before processing into paraffin blocks and sectioning at 5 µm. Cell counts of PCNA-positive cells were done manually in the DL and SDL using the 'Count' tool on Adobe Photoshop CC (RRID:SCR_014199), and statistical significance was evaluated with Student's *t*-test in Microsoft Excel (RRID:SCR_016137).

## Laser microdissection and RNA extraction

Jaws of bearded dragons at early postnatal stage (4 days post-hatching, corresponding to well-developed SDL in both pleurodont and acrodont teeth) were dissected in ice-cold phosphate buffered saline (PBS) and subsequently decalcified for 6 hr in 20% EDTA at 4°C with hourly changes of solution. Following decalcification, EDTA was washed out in PBS, and jaws were snap-frozen in liquid nitrogen and embedded in OCT compound (Tissue-Tek). Thick sections of 25 µm were prepared using a Leitz 1720 cryostat and mounted on Superfrost Ultra Plus slides (Thermo Scientific) at −25°C. Completed slides were dipped into ice-cold 70% ethanol and stored at −25°C while sectioning. For dental tissue visualization, slides were washed in ice-cold MilliQ water for 1–2 min, stained in cold 1% (w/v) cresyl violet solution in 50% ethanol for 10 s, and then washed in cold 70% and 100% ethanol for 1–2 min. After air-drying, the stained sections were stored at −80°C before use. Staining-guided laser microdissection was used to separate both the SDL and surrounding mesenchymal tissue from sections of both acrodont and pleurodont teeth, using a Zeiss PALM Microbeam device equipped with a Zeiss AxioCam IC camera and PALM Robo software (RRID:SCR_014435). As cresyl

violet staining allows epithelium to be well-distinguished, the SDL structure, defined as the free end of the DL epithelium that projects away from the tooth (see *Figure 5A*), was collected first under high magnification; mesenchymal tissue surrounding the now-removed SDL was then collected (see *Figure 5B*), thus ensuring precise tissue separation during collection. Tissue from multiple pleurodont and acrodont teeth from single individuals (*n* = 3) was pooled together to increase RNA yield. Total RNA was extracted using the RNeasy Plus Micro Kit (Qiagen), according to the manufacturer's instructions, and RNA integrity and concentration was assessed using Bioanalyzer RNA 6000 pico assays (Agilent).

## RNA sequencing and transcriptome analysis

Ribosomal RNA (rRNA) depleted libraries were produced using the Nugen Ovation SoLo kit, and paired-end sequencing was done on a NextSeq 500 platform (Illumina, U.S.A.). Quality control checks on raw sequence data were performed with the FastQC tool v0.11.8 (RRID:SCR_014583). Quality trimming, length filtering, and adapter sequence removal of reads were done using the Trimmomatic tool (RRID:SCR_011848; *Bolger et al., 2014*), and the SortMeRNA pipeline (RRID:SCR_014402; *Kopylova et al., 2012*) was used for additional rRNA filtering. The STAR software (RRID:SCR_015899; *Dobin et al., 2013*) was used for sequence alignment to the *Pogona vitticeps* genome (*Georges et al., 2015*) and for generating count data. Count data were subsequently used for normalization (*Robinson and Oshlack, 2010*), and differential gene expression analysis was conducted with edgeR (RRID:SCR_012802; *Robinson et al., 2010*), using three biological replicates. Genes with False Discovery Rate (FDR)-corrected p-values<0.05 were considered differentially expressed. The resulting list of genes was subjected to Gene Ontology (GO)-term functional annotation using the tool available on GO Consortium website (http://geneontology.org/), and genes from the GO category 'developmental process' were compared with vertebrate tooth gene expression (http://bite-it.helsinki.fi/) and mouse gene expression (http://www.informatics.jax.org/expression.shtml) databases.

## Quantitative PCR (qPCR)

Complementary DNA was generated by reverse transcription using the QuantiTect kit (Qiagen), according to the manufacturer's instructions, and analyzed by qPCR using iTaq Universal SYBR Green Supermix (Bio-Rad) and CFX96 real-time PCR detection system (Bio-Rad). Amplicons were designed from 80 to 95 bp in length with a melting temperature of 61°C, based on the publicly available *Pogona vitticeps* genome sequence (*Georges et al., 2015*). The housekeeping gene $\beta$-actin (*ACTB*) was identified as the best internal control for normalization. Each reaction was performed with four technical replicates using cDNA from three biological replicates. Results were analyzed with CFX Manager 3.1.1517.0823 software (RRID:SCR_017251) and Microsoft Excel (RRID:SCR_016137) using the double ΔCt method. The following primers were used: forward primer (fp) *ALX1*, 5'-GCAGTTCCGTTGTGACTTC-3'; reverse primer (rp) *ALX1*, 5'-ATCTGTCCGAGGTGAATGG-3'; fp *SIX3*, 5'-CTCTACCACATCCTGGAGAAC-3'; rp *SIX3*, 5'-TTCCTGGTAGTGAGCTTCG-3'; fp *ISL1*, 5'-TGCGGCAATCAAATCCAC-3'; rp *ISL1*, 5'-GGTTACATTCCGCACACTTC-3'; fp forkhead box I1 (*FOXI1*), 5'-GGCTATACTGGTTCAGTCCTC-3'; rp *FOXI1*, 5'-ACTTCAGTGCCCTCTCTTG-3'; fp BarH-like homeobox 1 (*BARX1*), 5'-AAGGTGGAGGGCTTGAATC-3'; rp *BARX1*, 5'-TGTCAACTGCTCGCTACTG-3'; fp *ACTB*, 5'-CCTGGAGAAGAGCTACGAAC-3'; rp *ACTB*, 5'-AGAAAGACGGCTGGAAGAG-3'.

## Acknowledgements

We thank Ann-Christine Aho, Maria Partanen, and Madara Snepere for technical assistance in captive breeding/animal care; Heikki Suhonen (University of Helsinki, Finland) for access to X-ray computed tomography facilities; Petri Auvinen and the DNA Sequencing and Genomics Laboratory (University of Helsinki, Finland) for access to sequencing facility; the Light Microscopy Unit (University of Helsinki, Finland) for access to imaging facility; the HiLAPS platform (University of Helsinki, Finland) for access to histology and laser microdissection facilities; Blanca Fernández López, Jacqueline Moustakas-Verho, Mona Christensen, Fabien Lafuma, Maria Sanz Navarro, Anamaria Balic, and Leah Biggs for assistance and technical advice; Marja Mikkola, David Rice, members of the Di-Poï laboratory, and Evo-Devo community (University of Helsinki, Finland) for helpful discussions.

## Additional information

### Funding

| Funder | Grant reference number | Author |
| --- | --- | --- |
| Suomen Akatemia | | Imran Khan<br>Nicolas Di-Poi |
| Helsingin Yliopisto | Integrative Life Science Doctoral Program | Lotta Salomies |
| Helsingin Yliopisto | | Nicolas Di-Poï |
| Biocentrum Helsinki | | Nicolas Di-Poï |
| Helsingin Yliopisto | Institute of Biotechnology | Nicolas Di-Poï |

The funders had no role in study design, data collection and interpretation, or the decision to submit the work for publication.

### Author contributions

Lotta Salomies, Conceptualization, Data curation, Formal analysis, Funding acquisition, Validation, Investigation, Visualization, Methodology, Writing—original draft, Writing—review and editing; Julia Eymann, Conceptualization, Data curation, Formal analysis, Validation, Investigation, Methodology, Writing—review and editing; Imran Khan, Formal analysis, Funding acquisition, Investigation, Methodology, Writing—review and editing; Nicolas Di-Poï, Conceptualization, Resources, Data curation, Supervision, Funding acquisition, Validation, Investigation, Methodology, Project administration, Writing—review and editing

### Author ORCIDs

Lotta Salomies http://orcid.org/0000-0003-1518-153X
Julia Eymann http://orcid.org/0000-0001-8147-9161
Nicolas Di-Poï https://orcid.org/0000-0002-3313-3016

### Ethics

Animal experimentation: All reptile captive breedings and experiments were approved by the Laboratory Animal Centre (LAC) of the University of Helsinki and/or the National Animal Experiment Board (ELLA) in Finland (license numbers ESLH-2007-07445/ym-23, ESAVI/7484/04.10.07/2016, and ESAVI/13139/04.10.05/2017).

### Decision letter and Author response

Decision letter https://doi.org/10.7554/eLife.47702.018
Author response https://doi.org/10.7554/eLife.47702.019

## Additional files

### Supplementary files

• Supplementary file 1. List of differentially expressed genes in transcriptomic analysis. List of significantly upregulated (UP) and downregulated (DOWN) genes in acrodont vs pleurodont SDL and acrodont vs pleurodont mesenchyme, with gene names, fold change (FC), *P*-values, and False Discovery Rate (FDR)-corrected *P*-values.
DOI: https://doi.org/10.7554/eLife.47702.013

• Transparent reporting form
DOI: https://doi.org/10.7554/eLife.47702.014

### Data availability

All Illumina reads have been deposited on Dryad Digital Repository under the link https://doi.org/10.5061/dryad.k66jn2s. Primers used for qPCR and ISH probes are available in the Key Resources

Table. All other data generated or analyzed during this study are included in the manuscript and Supplementary file 1.

The following dataset was generated:

| Author(s) | Year | Dataset title | Dataset URL | Database and Identifier |
|---|---|---|---|---|
| Salomies L, Eymann J, Khan I, Di-Poï N | 2019 | Data from: The alternative regenerative strategy of bearded dragon unveils the key processes underlying vertebrate tooth renewal | https://doi.org/10.5061/dryad.k66jn2s | Dryad Digital Repository, 10.5061/dryad.k66jn2s |

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
