## [Decision Letter]

Thank you for submitting your article "The alternative regenerative strategy of bearded dragon unveils the key processes underlying vertebrate tooth renewal" for consideration by *eLife*. Your article has been reviewed by three peer reviewers, one of whom is a member of our Board of Reviewing Editors, and the evaluation has been overseen by Diethard Tautz as the Senior Editor. The following individuals involved in review of your submission have agreed to reveal their identity: Vincent Laudet (Reviewer #3).

The reviewers have discussed the reviews with one another and the Reviewing Editor has drafted this decision to help you prepare a revised submission.

Summary:

The manuscript by Salomies et al., describes odontogenesis in bearded dragon with focus on differences in replacement ability of rostral and caudal teeth. All reviewers and the Reviewing Editor strongly agree that the bearded dragon represents a valuable model for comparative research as distinct replacement strategy can be observed in different areas of the jaw in one animal. Based on our communication, the reviewers and Reviewing Editor agree on the following essential revisions before this manuscript will be suitable for publishing in *eLife*.

Essential revisions:

1) The paper lacks a strong narrative story that would highlight the major results (such as the OE population, the difference between teeth types etc.) and better link the different manuscript components. We suggest that the authors add a paragraph to the Introduction or otherwise rework the Introduction to explain clearly what this model brings to the field and why it really stands out from what have been previously done to clearly highlighting the model's potential impact on the field.

2) We also suggest that authors better highlight and if possible, expand on the results of their transcriptomic analyses, which we feel are the most novel parts of the study. As it is, we are left to a small panel and our imagination to go beyond these data. After the basic characterization of the system depicted in Figure 1, we suggest the authors consider using these markers to better characterize the acrodont and pleurodont teeth in subsequent experiments rather than only using classical markers that are informative of course but obviously only partial. Specifically, we recommend the authors expand the ISH experiments shown in Figure 6 to include the expression of these markers in the mesenchymes of acrodont and pleurodont teeth.

- Materials and methods Section: Pictures or drawings of laser microdissected areas used for RNA extraction are not included. Could the authors please clarify which stage/age was used for this analysis?

- Figure 1: Rearrangement of acrodont teeth (E-J) into one row and corresponding number of pleurodont pictures into the second row of the plate would be helpful. Also labeling on the side would make figure more clear.

- We suggest that the authors present Figure 6 differently with a phylogenetic tree which also includes the situation in mouse and the fish situation. We realize the data are difficult to compare and that there will be many question marks, but it is the only informative way of presenting a model on such a question.

- On the transcriptomic studies we have come concerns on how to interpret the differential expression seen in the mesenchyme tissue. Could this be linked to a differential "contamination" level of the preparation by mesenchyme? It would be important to surely avoid any bias here, and we suggest that the authors address this concern in the manuscript.

- One reviewer strongly disagrees with the first sentence of the Abstract. According to this reviewer, "Zebrafish is, I think now a conventional model and as most teleost fishes is a polyphyodont species." We therefore suggest you edit this sentence.

- We suggest that the authors do not unite data coming from different animals, such as alligators and lizards, even though they might fall within a broader taxonomic group. We suggest the authors discuss all taxa separately.

---

## [Author Response]

Essential revisions:1) The paper lacks a strong narrative story that would highlight the major results (such as the OE population, the difference between teeth types etc.) and better link the different manuscript components. We suggest that the authors add a paragraph to the Introduction or otherwise rework the Introduction to explain clearly what this model brings to the field and why it really stands out from what have been previously done to clearly highlighting the model's potential impact on the field.

We thank the reviewers for this suggestion, and we now entirely revised the last paragraph of the Introduction to better highlight the various key features of the bearded dragon dentition (including new major results from our manuscript) as well as the potential impact of the emerging bearded dragon model in the tooth biology field. Particularly, this model organism offers a powerful system to elucidate the developmental and genetic basis of both evolutionary novelty and tooth regeneration in vertebrates. We also revised the last paragraph of the Discussion section to further emphasize our conclusions as well as the importance of this lizard model in answering both developmental and evolutionary questions.

2) We also suggest that authors better highlight and if possible, expand on the results of their transcriptomic analyses, which we feel are the most novel parts of the study. As it is, we are left to a small panel and our imagination to go beyond these data. After the basic characterization of the system depicted in Figure 1, we suggest the authors consider using these markers to better characterize the acrodont and pleurodont teeth in subsequent experiments rather than only using classical markers that are informative of course but obviously only partial. Specifically, we recommend the authors expand the ISH experiments shown in Figure 6 to include the expression of these markers in the mesenchymes of acrodont and pleurodont teeth.

We thank the reviewers for highlighting the importance of our new transcriptomic analysis. As suggested by the reviewers, we now expanded the ISH experiments by investigating and comparing the expression pattern of newly identified dental genes such as *ALX1, SIX3*, and *ISL1* in both acrodont and pleurodont teeth at various embryonic and postnatal developmental stages described in Figure 1 and Figure 2. Due to the amount of new results, all expression patterns (including data originally present in small panel E of Figure 5 mentioned by the reviewers) are now shown in a new Figure 6, and Figure 5 has been revised accordingly to only show the transcriptomic and quantitative PCR data. New magnified views of acrodont and pleurodont teeth at early postnatal stage (corresponding to the transcriptomic analysis) were also added to better characterize the expression patterns of these genes in both the successional dental lamina and mesenchyme regions. We also modified the Results section and Discussion section of the text accordingly to include these new important data in the revised manuscript. Particularly, our data strongly support our transcriptomic and PCR analyses, but also reveal dynamic expression pattern in epithelial and/or mesenchymal tissues starting from early tooth development. We really thank the reviewers for this suggestion, which further highlight the importance of the newly identified genes in odontogenesis and SDL patterning.

- Materials and methods Section: Pictures or drawings of laser microdissected areas used for RNA extraction are not included. Could the authors please clarify which stage/age was used for this analysis?

This is an understandable request. Drawings of laser microdissected areas used for RNA extraction and transcriptomic analysis are now included in the revised Figure 5A for both successional dental lamina (left panel) and dental mesenchyme (right panel), and the laser microdissection procedure is explained in more details in the Results section and Materials and methods section of the revised manuscript (see also comment below). An early postnatal stage (4 days post-hatching) was selected for this analysis in order to get a well-developed SDL in both pleurodont and acrodont teeth. We now clarified this issue in the Materials and methods section and in the Figure legends of the revised manuscript.

- Figure 1: Rearrangement of acrodont teeth (E-J) into one row and corresponding number of pleurodont pictures into the second row of the plate would be helpful. Also labeling on the side would make figure more clear.

We totally agree with this comment and have now modified the Figure 1 accordingly by first rearranging acrodont pictures in one row (panels E-J), and then adding new Hematoxylin and eosin (H and E)-stained sections of pleurodont teeth in a second row to get all corresponding stages (panels K-P). Labeling of tooth types is also now visible on the left side of the figure.

- We suggest that the authors present Figure 6 differently with a phylogenetic tree which also includes the situation in mouse and the fish situation. We realize the data are difficult to compare and that there will be many question marks, but it is the only informative way of presenting a model on such a question.

As mentioned by the reviewers, molecular findings from different species are rather difficult to compare, and data on stem cell characterization and particular gene expression pattern (including *SOX2*) are sometimes included in separate publications for a given species. However, we agree that this suggestion would help in the direct comparison of different vertebrate models, and we now revised the corresponding Figure (new Figure 7 of revised manuscript) by adding both a phylogenetic tree and new species from different vertebrate groups. We particularly focused on species (American alligator, mouse, pufferfish, cichlid fish, Atlantic salmon, African bichir) with comparable, integrative molecular data available for oral teeth, including mapping of slow-cycling putative stem cells and *SOX2* expression pattern. For some reptile species, the co-localization of *SOX2* expression with putative stem cells has not been directly assessed, as clearly mentioned in the revised Figure legend. New references on fish and mouse teeth have been also added at different locations in the revised manuscript.

- On the transcriptomic studies we have come concerns on how to interpret the differential expression seen in the mesenchyme tissue. Could this be linked to a differential "contamination" level of the preparation by mesenchyme? It would be important to surely avoid any bias here, and we suggest that the authors address this concern in the manuscript.

We agree that this could be a concern. However, to minimize any risk of “contamination”, all microdissections were guided by cresyl violet staining of dental tissue sections under high magnification, and the microdissection/camera device used enabled excellent visualization. As cresyl violet staining allows epithelium to be well-distinguished, the successional dental lamina (SDL) structure, defined as the free end of the dental lamina epithelium that projects away from the tooth (see revised Figure 5A and comment above), was collected first; mesenchymal tissue surrounding the now-removed SDL was then collected (see revised Figure 5A and comment above), thus ensuring precise tissue separation during collection. We are convinced that such method is more efficient than manual tissue dissection previously done in the field. As suggested by the reviewers (see also other comment above), we now further detailed the laser microdissection procedure in the Results section and Materials and methods section of the revised manuscript (with drawings of laser microdissected areas, see above).

- One reviewer strongly disagrees with the first sentence of the Abstract. According to this reviewer, "Zebrafish is, I think now a conventional model and as most teleost fishes is a polyphyodont species." We therefore suggest you edit this sentence.

We fully agree that the *zebrafish is a well-established model organism with polyphyodonty restricted to the pharyngeal* dentition, but our sentence was in fact referring to both “polyphyodonty” and “oral dentition”. However, as suggested by the reviewers, we now clarified the first sentence of the Abstract as *Xenopus laevis* also shows in fact a polyphyodont dentition on the upper jaws. Particularly, we now better emphasize the lack of lifelong replacement for oral dentition in most conventional models. We have also edited a similar sentence in the Discussion section (see first sentence).

- We suggest that the authors do not unite data coming from different animals, such as alligators and lizards, even though they might fall within a broader taxonomic group. We suggest the authors discuss all taxa separately.

We agree with this comment and have now edited the Introduction, Results section and Discussion section of the revised manuscript in order not to unite dental structures and/or specific data from different polyphyodont taxa like alligators and lizards.